# Structural basis of pausing during transcription initiation in mycobacterium tuberculosis

**Litao Zheng** ⓘ ✉ **& Ke Xu** ⓘ ✉

In bacteria, RNA polymerase (RNAP) often pauses during the early stages of transcription initiation. The structural basis for these transient pauses remains unclear. Here, we present cryo-electron microscopy (cryo-EM) structures of the paused initiation complex (PIC) and initiation complex (IC) of *Myco-bacterium tuberculosis* (*Mtb*), which include the RNAP core enzyme, the ECF σ factor σ^E, transcription factor CarD, promoter DNA, and nascent RNA. Our structures with pre-melted scaffolds reveal an intermediate at the 6−7 nt stage compatible with a paused-like intermediate, associated with steric hindrance between the emerging RNA and the σ3.2 region. This clash triggers a swivel of the RNAP structural module and scrunching of the transcription bubble. We also observe positional rearrangement of the σ4 domain, suggesting a poised pre-escape state. In addition, complementary reconstructions with fully matched DNA scaffolds (N-IC and N-PIC) support the physiological relevance of the captured intermediates. Together, our results support the existence of a mechanistic checkpoint during transcription initiation and suggest an RNA-induced model how RNAP conformational dynamics regulate early transcription.

Transcription initiation is the first and critical step in gene expression across all kingdoms of life[1–5]. Bacterial RNA polymerase (RNAP) employs a σ factor to recognize gene promoter regions, unwind double-stranded DNA (dsDNA) to form a transcription bubble, facilitate the synthesis of initial short RNA transcripts, and ultimately either assists in promoter escape or results in abortive transcription[5–7]. Abortive transcription frequently causes pausing and release of short RNA products through stressed initiation complex (IC) intermediates during the early stages of transcription initiation[8–10]. Although numerous single-molecule fluorescence resonance energy transfer (sm-FRET) studies and biochemical experiments have been used to study the dynamic process of transcription initiation[11–13], our understanding of initial transcription, especially pausing during transcription initiation, remains limited because of the transient nature of intermediates and molecular heterogeneity. In addition, the dynamic

nature of the initial intermediate complexes poses a challenge for structural definition. Consequently, these pausing states from transcription initiation have not been captured in high-resolution structures.

Extra-cytoplasmic function (ECF) σ factors are the largest class of alternative σ factors related to primary σ factors (σ^A), comprising only three conserved modules (σ2 domain, σ4 domain, and the functional σ3.2 region), in contrast to σ^A, which contains σ1.1 region, σ1.2 region, σ2 domain, σ3 domain, σ3.2 region, and σ4 domain[14–19]. The σ2 and σ4 domains recognize the DNA promoter −10 element and −35 element, respectively[20–22]. The σ3.2 region plays several crucial roles, including linking σ2 to σ4, inserting into the RNAP active-center cleft to initiate de novo transcription, adopting a helical conformation to pre-organize the template single-stranded DNA (ssDNA), and occupying the path of nascent RNA to mimic an RNA molecule for interaction with the

Shanghai Key Laboratory of Anesthesiology and Brain Functional Modulation, Clinical Research Center for Anesthesiology and Perioperative Medicine, Translational Research Institute of Brain and Brain-like Intelligence, Shanghai Fourth People's Hospital, School of Medicine, Tongji University, Shanghai, China. ✉e-mail: zlt22@tongji.edu.cn; kx2129@tongji.edu.cn

template ssDNA[20–23]. However, as RNA synthesis proceeds, the σ3.2 region must be displaced due to steric interactions with the 5′-end of nascent RNA[24–26]. This displacement imposes energy barriers that contribute to pausing and abortive initiation[23,27–31], which are subsequently accommodated by scrunching: RNAP pulls the unwound single-stranded template DNA inward, accommodating the additional unwound DNA by creating kinks in the transcription bubble[28,29,32]. Scrunching allows RNAP to capture free energy from DNA stacking. This energy is subsequently stored and used to disrupt RNAP-promoter interactions, enabling promoter escape[28,29]. In summary, we selected the ECF σ[E] for structural analysis because it has a simpler domain organization (comprising σ2, σ3.2, and σ4) than the primary σ[A]. This minimal architecture made it experimentally more tractable for reconstitution and structural determination, enabling us to better capture initiation intermediates, including the paused complex.

The *Mtb* σ[E] factor is a critical ECF σ factor that mediates responses to the cell-surface and other extra-cytoplasmic stresses[5,14,33–39]. Although σ[E] shares high sequence homology with σ[H] and is predicted by AlphaFold to adopt a highly similar fold, direct experimental structural information for *Mtb* σ[E] factor is still limited. Recently Yu Zhang and Richard H. Ebright groups have reported the structures of *Mtb* transcription initiation complexes comprising the ECF σ factors σ[H] and σ[L], respectively[40,41]. σ[H] and σ[L] adopt homologous binding modes to the RNAP core and promoter DNA, recapitulating the σ-dependent holoenzyme architecture observed in bacterial transcription initiation[25,40,41]. These structures also revealed that the σ3.2 linker of ECF σ factor is inserted into the RNA exit channel and active center cleft, playing an essential role in open promoter complex (RPo) formation, promoter escape and abortive production[25,33,40,41]. In addition to σ[E], CarD is a transcriptional activator found widely across bacterial species, including *Mtb*, but is absent in *E. coli*[42,43]. The structure of CarD consists of an N-terminal domain that folds into a Tudor-like structure, which is similar to the *E. coli* transcription repair coupling factor, and serves as the RNAP-interacting domain and a C-terminal helical domain[42,43]. In *Mtb*, CarD has been described as a regulator of the principal σ subunit, particularly in ribosomal gene transcription[44], whereas its cooperation with ECF σ factors has not been reported. Notably, CarD has been reported to modulate transcription driven by an ECF σ factor in *Myxococcus xanthus*, suggesting that CarD may also influence σ factor–dependent transcription in *Mtb*, although this has not been experimentally characterized[45]. Thus, while the general mechanism by which ECF σ factors initiate transcription is increasingly well understood, the basis of transcriptional pausing during initiation remains poorly defined, in part due to limitations of traditional crystallographic approaches, and the potential involvement of CarD in this process has yet to be elucidated.

Here, we investigated the mechanism of pausing during transcription initiation by determining a series of cryo-electron microscopy (cryo-EM) structures of transcription complexes from *Mtb*, comprising the RNAP core enzyme, DNA/RNA scaffolds, ECF σ factor σ[E], and general transcription factor CarD. These complexes include (1) an initiation complex with 6-nt RNA (IC6), (2) an initiation complex with 7-nt RNA (IC7), and (3) a paused initiation complex with 7-nt RNA (PIC7). Our structures are consistent with RNA polymerase pausing during the transition between 6 and 7 nt transcripts, adopting a half-translocated conformation that swivels the RNAP, resulting in a 3-base pair unwinding and scrunching of the transcription bubble DNA. We also found that CarD binds to the template ssDNA in the IC to help maintain the transcription bubble, and interacts with the non-template ssDNA in the PIC to stabilize the bubble and potentially facilitate escape from the paused state. This difference in binding modes indicates that CarD plays an important role not only in the formation of the transcription bubble but also in the regulation of pausing during transcription initiation. In addition to these high-resolution structures

obtained with pre-melted scaffolds, we also reconstructed initiation complexes using fully complementary DNA scaffolds to assess physiological relevance. These include (4) an initiation complex with 6-nt RNA (N-IC6), (5) an initiation complex with 7-nt RNA (N-IC7), and (6) a paused initiation complex with 7-nt RNA (N-PIC7). Although resolved at lower resolution and without atomic modeling, these reconstructions are consistent with the pre-melted scaffold structures and support the validity of our scaffold design strategy. By combining analyses of these structures with biochemical experiments, we support the presence of a pausing checkpoint during transcription initiation, and provide an RNA-induced model to elucidate the mechanism underlying paused transcription initiation.

## Results

### Preparation of the PIC using a nucleic acid scaffold with a 6- or 7-nt RNA

The ICs containing *Mtb* ECF σ factors have been successfully prepared using nucleic acid scaffolds for structural studies[40]. According to kinetic analyses of RNAP transcription initiation, RNAP exit kinetics from initiation complexes stalled specifically at the 7-nt RNA transcript stage demonstrate delayed transition to elongation[11]. Additional research has also shown that the pause is encountered by RNAP after the synthesis of a 6-nt RNA[12]. To test whether *Mtb* RNAP dwells at this stage, we performed in vitro transcription assays using fully complementary DNA-RNA scaffolds. RNAP showed accumulation of transcripts at 6–7 nt, consistent with increased dwell time during extension from 6 to 7 nt, and supporting the idea that the RNA lengths used for our structural studies correspond to relevant early-initiation intermediates (Supplementary Fig. 1H, I). These biochemical studies are therefore consistent with RNAP dwelling at the 6–7 nt stage, in agreement with the structural capture of PIC-like intermediates. In these assays, we use the fully matched scaffold to relate the 6–7 nt products directly to the cryo-EM architecture, whereas pre-melted scaffolds are interpreted as qualitative support for the presence of a 6–7 nt checkpoint.

Based on precedents from structural studies of mycobacterial RNAP and ECF σ factors (Supplementary Table 1), we adopted pre-melted nucleic acid scaffolds for IC6, IC7, and PIC7, as these designs are widely used to stabilize transcription initiation intermediates beyond RPo and allow visualization of the interactions between nascent RNA and the σ3.2 region. To investigate the potential formation of PIC and the mechanism of pausing, we prepared transcription initiation complexes that contained RNAP-core, σ[E] factor, CarD, and nucleic acid scaffolds with a 6- or 7-nt RNA. We tested the binding of σ[E] factor and CarD to the RNAP-core using native electrophoretic mobility shift assays (EMSA), and found that adding σ[E] factor and CarD to the RNAP core pre-incubated with DNA/RNA scaffold resulted in an upward shift of the band, indicating the formation of a potential complex (Supplementary Fig. 1G). After collecting and processing 9518 cryo-EM micrograph movies in the transcription initiation complex with a 7-nt RNA, two major classes were identified: 55% of the particles belonged to IC, and 45% to PIC (Supplementary Fig. 2). This distribution likely reflects the relative stability and population of these two conformations under the experimental conditions, representing alternative paused states within the initiation checkpoint. Additional transient intermediates may exist but were not captured due to their low population and dynamic nature; these could include partially scrunched states, transiently backtracked complexes, or short-lived RNA-σ3.2 interactions, consistent with the idea that cryo-EM preferentially samples the most stable states. At a matched resolution of 3.3 Å, the IC7 reconstruction utilized 75,465 particles while the PIC7 employed 65,908 particles. (Supplementary Fig. 2 and Supplementary Table 2). Only the major class was identified by collecting and processing 2786 cryo-EM micrograph movies in the transcription initiation complex

with a 6-nt RNA which belonged to the IC. At a resolution of 3.5 Å, the IC6 reconstruction employed 65,079 particles following the same processing workflow (Supplementary Fig. 3 and Supplementary Table 2). All transcription complexes showed high-quality densities for RNAP, the σ2 domain of σ$^E$, and DNA/RNA scaffold, except for the σ4 domain of σ$^E$ and CarD, whose densities were relatively weak but still sufficient to position the main chain (Supplementary Fig. 2, 3).

In addition to these high-resolution structures obtained with pre-melted scaffolds, we also performed reconstructions using fully complementary DNA scaffolds to assess physiological relevance. These reconstructions yielded N-IC6, N-IC7, and N-PIC7 (Supplementary Figs. 4, 5 and Supplementary Table 3). Although the maps were at lower resolution and lacked continuous density for parts of the DNA/RNA scaffold, they nevertheless showed overall conformational features consistent with the corresponding pre-melted scaffold structures (Supplementary Fig. 4C). Because of the limited resolution and incomplete nucleic acid density, atomic models could not be built for these reconstructions. These results support the rationale of using pre-melted scaffolds and are consistent with the view that the captured IC and PIC states represent physiologically relevant intermediates. We therefore focused subsequent structural analyses on the high-resolution IC6, IC7, and PIC7 reconstructions, while using N-IC6, N-IC7, and N-PIC7 as supportive validation datasets.

## Overall structure of IC and PIC

In the IC structure, which includes IC6 and IC7 that are nearly identical except for RNA length differences and σ3.2 tip conformational changes, CarD and σ$^E$ maintain equivalent interactions with RNAP as those observed in other ECF σ factor structures[40,41]. CarD forms a minor groove wedge to stabilize template ssDNA in the transcription bubble (Fig. 1A, B). The σ2 domain of σ$^E$ engages the clamp module of RNAP β′ subunit via a broad interaction interface (Fig. 1B). In contrast, the σ4 domain of σ$^E$ encapsulates the tip helix of the RNAP β flap domain (βFTH) within a tightly fitting structural pocket (Fig. 1B). The structure of the IC also revealed multiple interactions in promoter recognition and promoter unwinding by σ$^E$. The σ4 domain of σ$^E$ anchors to the major groove of the upstream DNA and reads the sequence of the −35 element of the upstream dsDNA (Fig. 1B and Supplementary Fig. 6A). The σ2 domain of σ$^E$ interacts with the junction and fork of the-10 element of the upstream ds/ssDNA to unwind the promoter DNA, similar to other ECF factors (Fig. 1B and Supplementary Fig. 6A)[25,40,41]. Moreover, the σ3.2 region of σ$^E$ inserts into the active center cleft to guide the template ssDNA into the RNAP active site (Fig. 1B). The DNA/RNA hybrid base pairs remain in a post-translocation state within the active site, with a root-mean-square deviation (RMSD) of 1.819 A° for all Ca atoms compared to the post-translocation state in Mtb (PDB:5ZX2) (Fig. 1C, D and Supplementary Fig. 7A, B).

In the PIC structure, while the interactions of the σ2 and σ4 domains of σ$^E$ with RNAP resemble those in the IC, multiple conformational changes differing from those observed in the IC are evident (Fig. 1A, E). Unlike the structure of IC, the PIC exhibited a half-translocated DNA/RNA hybrid, with an RMSD of 0.966 Å for all Ca atoms compared to the half-translocation state in Mtb (PDB:8E8M) (Fig. 1F, G and Supplementary Fig. 7C, D). The swivel module (defined in Supplementary Table 4) rotated relative to the rest of the RNAP, resulting in rotation of the promoter upstream dsDNA (Fig. 1B, E). The upstream ds/ssDNA junction also undergoes unwinding of three bases, which may be caused by the combined effects of RNAP swiveling and the raising of the upstream dsDNA (Fig. 1B, E and Supplementary Fig. 6b). Moreover, the σ4 domain of σ$^E$, which anchors the major groove of the upstream dsDNA, shows a significant conformational change in PIC compared to IC (Fig. 1B, E). Surprisingly, CarD also formed a minor groove wedge but stabilized the non-template ssDNA in the transcription bubble (Fig. 1B, E).

## The conflict between the σ3.2 and nascent RNA induces pausing of RNAP

The σ3.2 region of σ$^E$ inserts into the active center cleft and occupies the RNA exit channel, which means it must be displaced as RNA elongates[24–26]. It has been reported that RNAP adopts a pausing state when the nascent RNA extends to 6 to 7 nucleotides, with the first evidence demonstrated in 2016[11–13]. However, conformational changes in RNAP during pausing transcription initiation have not been directly observed. Additionally, RNAP swiveling, as a conserved feature, has been reported in previous studies on Mtb NusG-dependent pausing and the E. coli pause hairpin (PH) stabilized pausing elongation complex[46–48]. Despite these findings, it remains unknown whether RNAP swiveling occurs during pausing transcription initiation, what drives the change in the RNAP active site, and how the σ3.2 region of σ$^E$ is displaced.

Structural evidence from PIC7 provides key insights into these questions. The structure of PIC7 revealed that Mtb RNAP undergoes swiveling during pausing transcription initiation (Fig. 2A). Swiveling associated with pausing transcription initiation is similar to Mtb NusG-dependent pausing, which is involved in transcription elongation and termination, suggesting that swiveling may be a conserved characteristic in all pausing processes of transcription[46–48]. The PIC was swiveled and half-translocated, whereas the IC was unswiveled and post-translocated (Fig. 2A and Supplementary Fig. 7A–D). In the half-translocation state, the RNA moves forward, but the template DNA is not completely translocated. The swiveling of RNAP also distorts the bridge helix (BH), which is consistent with the known ability of Mtb NusG to facilitate pausing through interaction with the conserved TTNTTT motif[46–48]. The C-terminal portion of BH was attached to the swivel module, whereas the N-terminal portion of BH was attached to the unswivel module of the RNAP (Fig. 2B). Therefore, the middle of BH is kinked and the C-terminal portion of BH is shifted toward the template DNA, resulting in steric clash with the +1 position of template DNA in the nucleotide addition cycle (Fig. 2B).

Central to pausing maintenance is the σ3.2 region's obstruction of RNA extension. The nascent RNA encountered a roadblock formed by the σ3.2 region of σ$^E$, which drives RNAP swiveling and maintains the pausing state. Although the swiveling associated with pausing is a conserved characteristic in all pausing processes of transcription, the reasons for driving swiveling in each pausing process are distinct. The E. coli his RNA hairpin stabilizes pausing through its interaction with the RNA exit channel, while Mtb NusG facilitates pro-pausing through the interaction of its N-terminal NGN domain with a conserved TTNTTT motif[46–48]. However, in Mtb σ$^E$-dependent pausing, the σ3.2 region of σ$^E$ inserts into the RNA exit channel, forming a steric roadblock that clashes with the extending RNA. This steric interference promotes RNAP swiveling and contributes to transcriptional pausing, as supported by our in vitro transcription assays (Fig. 2B–E, Supplementary Fig. 1H–K), which reveal that pausing occurs during the transition from 6-nt to 7-nt RNA synthesis. This behavior is consistent with previous observations in the E. coli σ$^{70}$-RNAP system[11].

Conformational restructuring of σ$^E$ further elucidates pausing regulation. In the structures of IC6 and IC7, the σ2 and σ4 domains of σ$^E$ are located on the surface formed by RNAP, while the σ3.2 region of σ$^E$, which likely coincides with the movement of the 5′-end of RNA and causes steric clash between the nascent RNA and the tip of the σ3.2 region, is inserted into the RNA exit channel (Fig. 2C–E). In the structure of PIC7, the folding of σ2 of σ$^E$ is virtually identical to that in IC6 and IC7, while the σ4 domain of σ$^E$, which binds to the major groove of the upstream dsDNA, exhibits a remarkable conformational change within its relative position (Supplementary Fig. 8A–C). The σ4 domain in PIC7 rotates -55° relative to its position in IC6, moving from the RNA exit channel region toward the upstream dsDNA. (Fig. 1B, E and Supplementary Fig. 8A). In IC and other ECF σ complexes, the σ4 central region is positioned 10 Å from the RNA exit channel[25,40,41] (Fig. 1B, E). Upon transition to the PIC, σ4 central region undergoes rotation to a

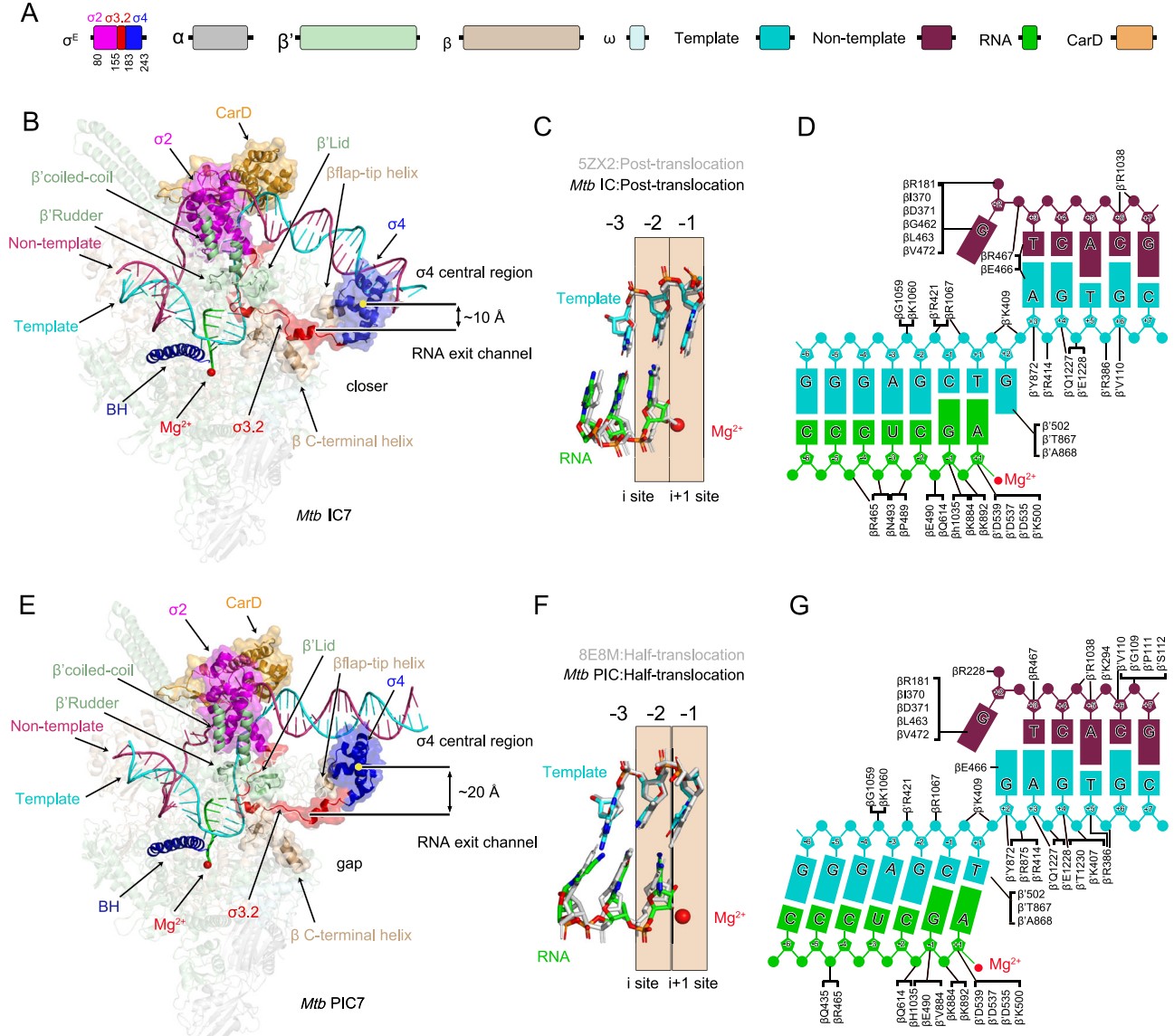

**Fig. 1 | Overall structure of the complex. A** Structural organization of *Mtb* σ^E and the component in σ^E_CarD_RNAP complex. Conserved regions σ2, σ3.2 linker and σ4 are in magenta, red, and blue, respectively. The color scheme for α, β, β', ω, template, non- template, RNA, and CarD are in gray, pale green, wheat, pale cyan, cyan, warm pink, green, and bright orange, respectively. **B,E** Protein-protein interactions by σ^E factor in *Mtb* IC (**B**) and PIC (**E**). The α, β, β', and ω subunits of RNA polymerase core enzyme is shown as cartoon (50% transparency). The nontemplate DNA, template DNA, RNA strands, σ^E and CarD are shown as surface. The σ4 central region and the catalytic Mg²⁺ ion are depicted as yellow and red spheres, respectively. **C,F** RNAP translocation states are compared to the IC and PIC in the middle, with reference structures (PDB 5ZX2, 8E8M) colored gray. The post-translocation state in IC (**C**), and the Half-translocation state in PIC (**F**). The DNA/RNA residues are shown as colored sticks. **D, G.** Protein-nucleic acid interactions in downstream dsDNA and RNAP activity center are indicated on the right.

position 20 Å away from the channel, driving dissociation of the σ3.2 region from the β C-terminal helix (Fig. 1B, E). This previously unobserved position of the σ4 domain of σ^E not only causes variation in the σ^E and RNAP interactions, but also leads to significant differences in σ^E and DNA interactions (Supplementary Fig. 6A, B). It is worth noting that despite low sequence conservation in the σ3.2 region across ECF σ factors, we observe striking structural conservation of their flexible linker conformations (Supplementary Fig. 8A, D). This suggests a convergent functional mechanism for transcriptional pausing regulation, potentially mediated through σ3.2-dependent steric clashes during RNA extension.

## The upstream DNA scrunching, unwinding, and raising in PIC
Before the promoter escapes, RNAP must provide sufficient time for DNA scrunching to accumulate the necessary free energy[28–30]. The highly stable scrunched state and extensive pausing during transcription initiation have been observed using sm-FRET[11,12,25]; however, the structure of pausing during transcription initiation, involving a scrunched, unwound, and raised state, has not been obtained. Our structural determination of the PIC reveals a 3-bp unwound region at the upstream ds/ssDNA junction, contrasting with the fully duplexed state observed in the IC (Fig. 3A, B). The driving force of unwinding may come from the compression of nascent RNA against the σ3.2 region, leading to the lifting of the σ4 domain and the upstream dsDNA, which in turn further promotes DNA unwinding (Fig. 3A–C). The unwinding in PIC causes the scrunching of DNA in the transcription bubble, in which the upstream dsDNA rotates approximately 35° around the ds/ssDNA junction relative to its position in the IC structure, resulting in the −35 element of the upstream dsDNA being positioned at an increased distance of approximately 20 Å from the RNA

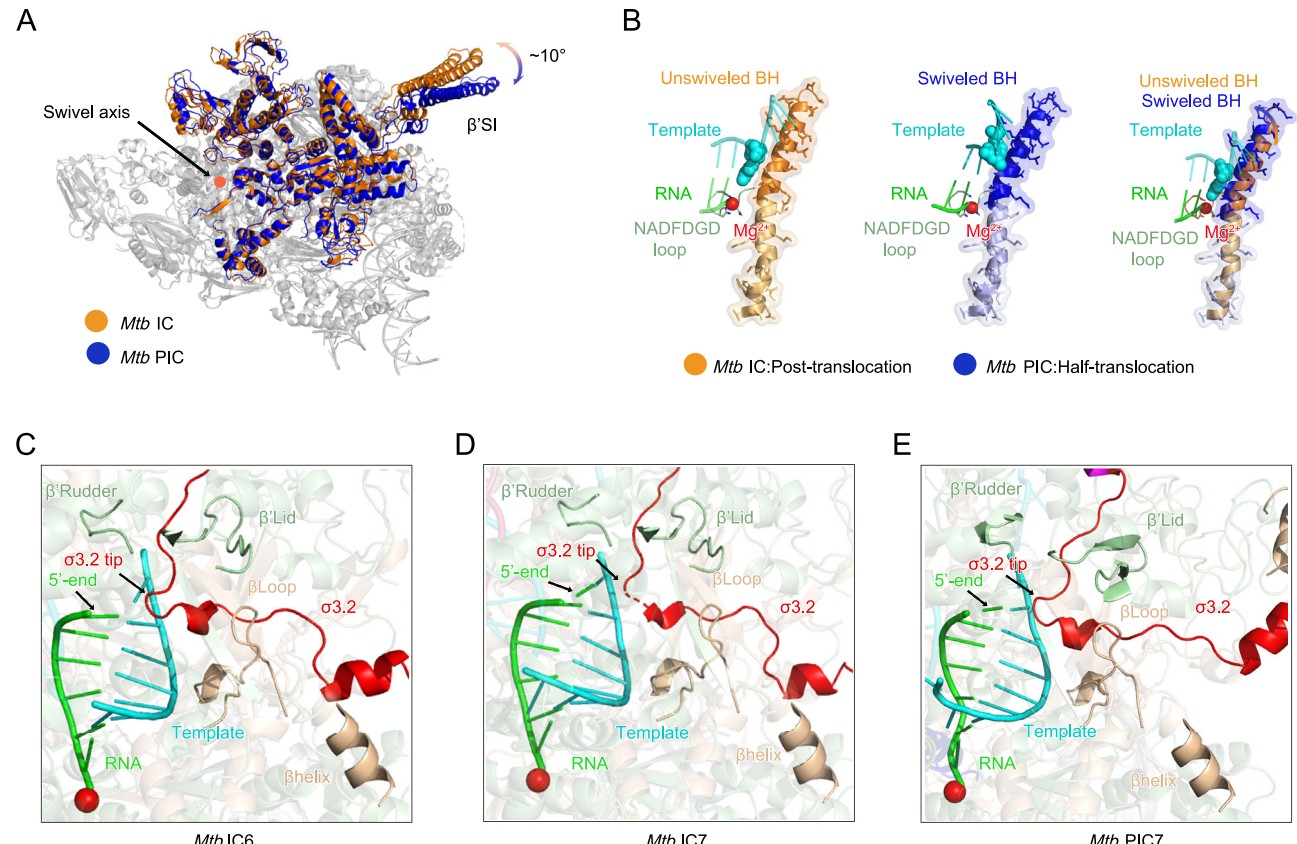

**Fig. 2 | Structural reorganization and steric competition during transcription initiation. A** Structural alignment through RNAP core domain (gray), reveals -10° clockwise rotation of the PIC swivel domain (blue) relative to IC (bright orange). Cartoon representations with 50% transparency. **B** Catalytic element dynamics. Superposition of bridge helix (BH: IC as bright orange, PIC as blue) and NADFDGD loop (pale green) highlights C-terminal BH reorientation toward DNA/RNA hybrid. The template DNA nucleotide in the i + 1 site and the catalytic $Mg^{2+}$ ion are depicted as cyan and red spheres, respectively. **C–E** RNA-driven displacement of the σ3.2 tip, showing structures of IC6 (**C**), IC7 (**D**), and PIC7 (**E**). Colors as in Fig. 1.

exit channel (Fig. 1E). Although the pre-melted DNA scaffold used in our study facilitates the stabilization of transcription intermediates suitable for structural determination, it may result in a lower degree of scrunching compared with that occurring during the actual transcription reaction. The true scrunched states are likely to be more compact and dynamic, but their intrinsic instability currently prevents high-resolution structural characterization. Nevertheless, the paused conformation we observe here is consistent with that seen using fully matched DNA templates, supporting that the captured structure faithfully reflects the pausing state of the transcription complex (Supplementary Figs. 4, 5). Although the DNA duplex undergoes unwinding and rotational adjustments in PIC, extensive interactions persist between the PIC-associated DNA/RNA scaffold and the RNAP-core. However, the spatial arrangement of these interaction sites differs significantly compared to those observed in the IC (Supplementary Fig. 6A, B). In PIC assembly, the σ4 domain of σ^E and the major groove of dsDNA of the −35 element are adjacently located, unlike the tight binding observed in the IC, but they still separate and form a clamp of a narrow cleft with a width of approximately 10 Å (Supplementary Fig. 8B, C). The σ4 domain helix spanning from K221 to L240 engages the −35 element through weak van der Waals interactions, with the closest contact of approximately 5 Å localized between the helical surface and the T-26 to T-28 bases (Fig. 3C, D and Supplementary Fig. 8B, C). This minimal interface suggests a transient association mechanism at the transcriptional checkpoint, i.e., the stage where RNAP either pauses or escapes into productive elongation, distinct from stable binding. The reduced σ^E-DNA binding and transcriptional activity observed with R226A, R228A, and R231A

substitutions could involve impaired recognition of the −35 element by the σ4 domain, along with modified structural rearrangements associated with σ4 release from the RNAP-core (Fig. 3E, F and Supplementary Fig. 8E). Thus, the IC and PIC represent two alternative conformations of RNAP at the 7-nt RNA stage, corresponding to distinct states within a transcriptional checkpoint.

## CarD stabilizes distinct ssDNA in both IC and PIC

CarD interacts directly with the RNAP β subunit β1-lobe and flanks one side of the upstream ds/ssDNA junction, facing the β' coiled-coil and the σ2 domain[42]. In both IC and PIC structures, CarD exhibits a significant reconfiguration in DNA-binding interactions, despite maintaining virtually identical folding with an all-Cα RMSD of 1.826 Å (Fig. 4A–E). In the structure of IC, the C-terminal helical domain of CarD directly interacts with the template DNA through two α-helices, using the hydrophobic planar side chain of W85 as a bulky wedge inserted into the minor groove at the junction of the template ssDNA (Fig. 4A, B), which is identical to that in previously reported CarD-RPo complexes[42]. The two α-helices were oriented approximately perpendicular to the dsDNA axis, establishing a relatively modest interaction interface of 380 Å² between CarD and the template ssDNA (Fig. 4A, B and Supplementary Fig. 9A–C). In the PIC structure, the C-terminal helical domain of CarD directly engages the non-template ssDNA via two α-helices (Fig. 4C, D). This interaction is stabilized by potential hydrogen bonds formed between the side chains of Y89 and R126 with the phosphate groups of the DNA backbone, creating an interface between CarD and the ssDNA that resembles the template-strand interaction observed in the IC (Fig. 4C, D). CarD flanks one side of the

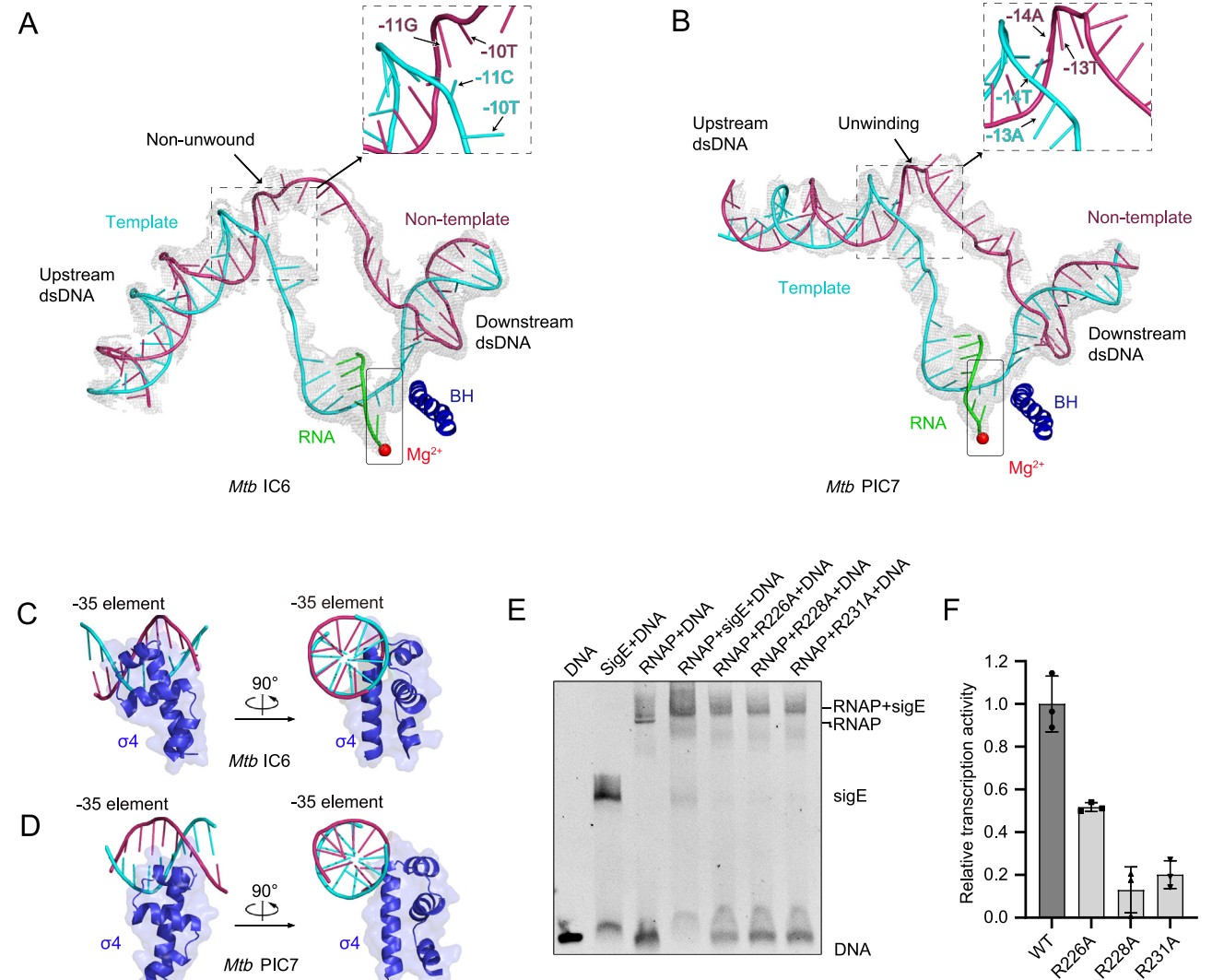

**Fig. 3 | DNA scaffold reorganization in PIC. A,B** The RNA/DNA hybrid scaffold is visualized with its corresponding cryo-EM density map (transparent gray mesh), highlighting structural integration within the transcriptional complex. **C,D** σ4 domain engagement with promoter −35 element. IC6 (**C**) and PIC7 (**D**) structures shown in dual orientations (front and side views), which also show different modes of combination. **E,F** Functional validation of interaction motifs. EMSA quantifies DNA binding affinity changes caused by σ4 point mutations and their effects on complex formation using the DNA substrate shown in Supplementary Fig. 1H (**E**). The *Mango*-fluorescence transcription assay reveals σ4-dependent transcriptional activation (**F**). The data are presented as mean ± S.E.M. $n = 3$ independent experiments. Source data are provided as a Source Data file.

non-template ssDNA, while the σ2 domain flanks the opposite side, together forming a tighter interaction with the non-template ssDNA to further stabilize the scrunched transcription bubble (Fig. 4C, D and Supplementary Fig. 9D–F). This stabilization likely prevents premature collapse of the bubble, while allowing the conformational flexibility required for RNAP to progress through the paused intermediate. Surprisingly, structural comparison reveals that the PIC-specific ordering of the σ2 N-terminal helix correlates with CarD positioning near σ2, suggesting that CarD stabilizes this region via direct spatial interactions (Fig. 4F). Consistent with these structural observations, our in vitro transcription assays showed that CarD enhances σ^E-dependent transcription and alters the distribution of RNA products at the 6−7 nt stage (Supplementary Fig. 1H−K), suggesting that CarD may play a dual role by stabilizing RNAP-σ-DNA complexes and potentially facilitating the transition from the paused to the elongating state during transcription initiation. Moreover, a time-resolved fluorescence assay using a Cy3-labeled promoter bubble was performed to assess the effect of CarD on open complex formation. In the presence of CarD, the fluorescence signal reached a higher plateau compared with the RNAP-σ^E-only complex, indicating that CarD stabilizes the σ^E-RPo complex and enhances its steady-state accumulation (Fig. 4G). These results are consistent with previous findings showing that CarD promotes the stabilization of transcription initiation complexes[49,50]. W85A, Y89A, and R126A mutations disrupted critical interactions, leading to a marked reduction in transcriptional activity (Fig. 4H). The structural studies of *Mtb* CarD-IC and CarD-PIC complexes presented here suggest that CarD plays an important role in forming the transcription bubble in the IC and in stabilizing the scrunched transcription bubble in the PIC.

## Discussion

Transcription pausing is a crucial mechanism found among RNA polymerases in both eukaryotic and prokaryotic cells[51–53]. While the assembly of transcription initiation complexes and NusG-dependent pausing of elongation complexes have been characterized in *Mtb*[46,48], the mechanism underlying pausing during transcription initiation remains unclear. Here, we reconstituted the *Mtb* PIC and determined its structure, providing insights into conformational changes during

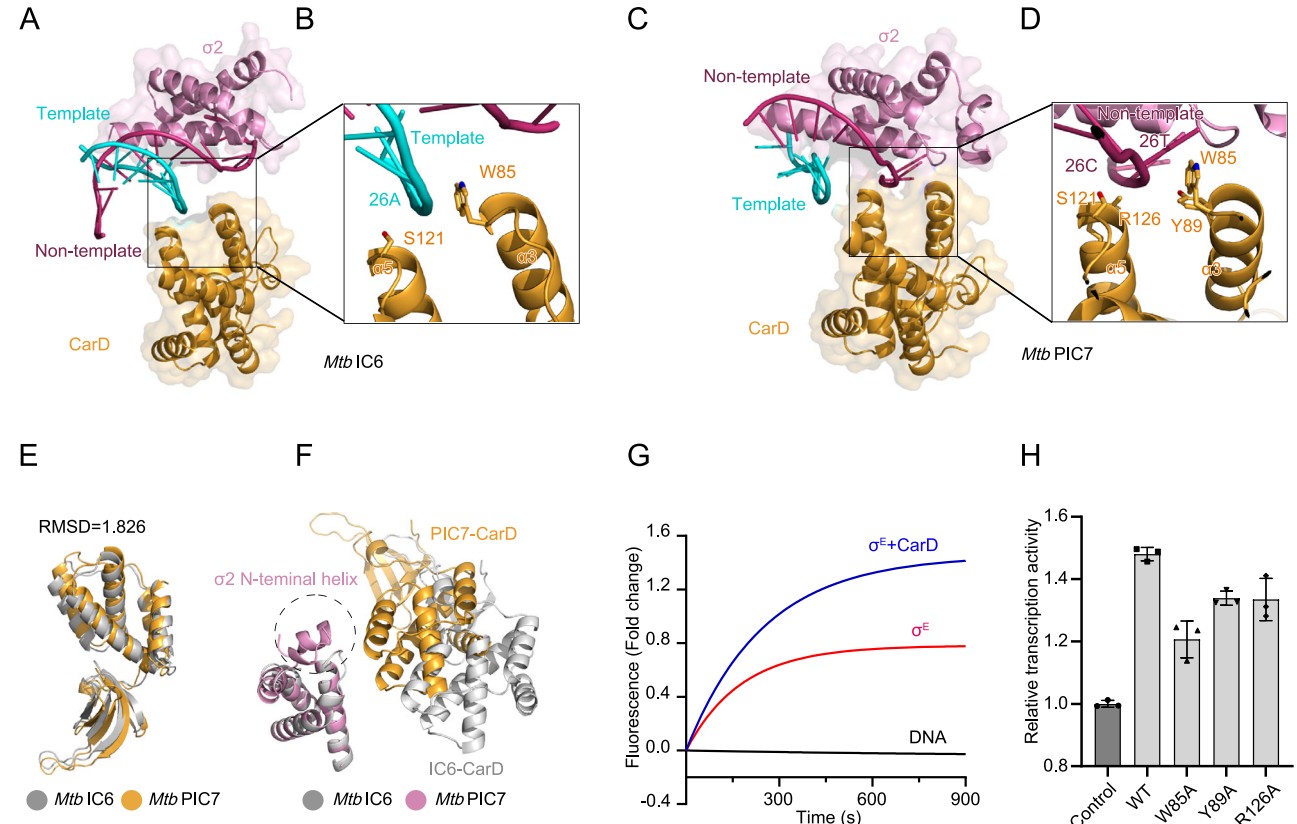

**Fig. 4 | Characterization of CarD binding motifs to ssDNA. A–D** Structural basis of CarD/σ2-DNA interactions. Front views of CarD and σ2 domain bound to template DNA in the IC (**A,B**). Corresponding interactions with non-template DNA in the PIC (**C,D**). Critical interaction regions (dashed boxes) are magnified in right panels. The CarD and σ2 domain are shown as cartoon and surface (50% transparency). DNA chains are shown as cartoon representations with color coding consistent with Fig. 1. **E** Structural superposition of IC6-CarD (gray) and PIC7-CarD (orange) reveals identical folding topology. **F** The structural superposition of IC6-σ2 (gray) and PIC7-σ2 (magenta) highlights the exclusive formation of an N-terminal helix, which is stabilized by the closer-positioned CarD in the PIC. **G** Time-resolved fluorescence assay assessing the ability of CarD to stabilize the RPo complex. Fluorescence signals were recorded every 3 s for 900 s and expressed as fold-change relative to DNA-only control. Data were fitted to a single-exponential association model $F_t = F_0 + A(1 - e^{-kt})$. The DNA scaffolds used in this assay are listed in Supplementary Table 7. Data are presented as mean ± S.E.M. ($K_{σE+CarD} = 0.00447 ± 0.00027$; $K_{σE} = 0.00655 ± 0.00045$; $A_{σE+CarD} = 1.461 ± 0.026$; $A_{σE} = 0.7675 ± 0.052$; n = 3 independent experiments). Source data are provided as a Source Data file. **H** Functional validation of interaction motifs. *Mango*-fluorescence assay reveals CarD-dependent cooperativity in transcriptional activation. The σE only condition was included as a control to assess baseline transcriptional activity. The data are presented as mean ± S.E.M. n = 3 independent experiments. Source data are provided as a Source Data file.

pausing. While the σ3.2 region is well established as a pausing determinant, our structures further reveal that it also facilitates RNAP swiveling through contacts with the 5′ end of the nascent RNA and the RNAP active center cleft. Consistent with prior biochemical data, the PIC structure shows a scrunched transcription bubble, capturing energy for promoter escape or RNA release[11–13,28]. Although the pre-melted scaffolds used in this study may exhibit a reduced degree of DNA scrunching compared with fully transcribing complexes, they provided a stable system that allowed us to capture otherwise transient paused intermediates at high resolution. This approach inevitably simplifies the transcription bubble dynamics, and the scrunched state resolved here may represent a less compact conformation than that formed during active transcription. However, the paused intermediate observed under these conditions showed the same key structural features as those obtained using fully matched DNA templates in this study (Supplementary Figs. 4, 5), supporting the view that the pre-melted design does not substantially alter the key features of RNAP rotation and pausing.

**An RNA-induced mechanism for pausing during transcription initiation**

Transcription initiation is a multistep and dynamic process carried out by an RNAP consisting of RNAP-core enzyme and a σ factor. The dsDNA promoter binds the main channel of the RNAP-core enzyme to form an RPo complex with an ~12-14 bp unwound transcription bubble[26,54]. In contrast, the main channel is closed when it binds a σ factor to form a holoenzyme that synthesizes initial short RNA transcripts[20]. RNAP is also capable of clamp opening and closing, a conserved motion observed in many cellular RNAPs[27,53,55,56], although DNA opening during initiation does not obligatorily require this conformational change[57]. Another conserved motion of RNAP is swiveling, which is associated with transcriptional pausing and has been observed in bacterial RNAPs[9,47,58–60]. Swiveling of the swivel module typically results in a slight degree (1 - 4°) of rotation relative to the rest of the module of the RNAP, using a rotation axis approximately parallel to the BH[60–62]. Recent studies have shown that *Mtb* RNAP employs a swiveling mechanism to maintain a pausing state during transcription elongation, with the universal transcription factor NusG acting as a pro-pausing factor[46,48]. Additionally, the *Mtb* RNAP swivel module is directly linked to TL folding, as NusG interacts with the TTNTTT motif of non-template ssDNA, promoting the rotation of the swivel module and thereby inhibiting TL folding and nucleotide addition[46,48]. In this study, we found that the swivel module of *Mtb* RNAP undergoes rotation, which may further stabilize the scrunched transcription bubble between the DNA/RNA hybrid and upstream dsDNA, and appears to be linked to upstream dsDNA unwinding (Fig. 3A, B).

Despite the absence of the NusG factor and the TTNTTT motif in non-template ssDNA within the PIC structure, swiveling of the *Mtb* RNAP swivel module still occurs, leading to BH distortion and pausing during transcription initiation, and this swiveling is stimulated by the σ3.2 region of the σ factor. Previous studies of initiation complexes, including Li et al., 2020 (X-ray structures)[24] and Gao et al., 2023 (RNAP-σ[54])[63], only observed steric clashes between the nascent RNA and the σ finger without detecting swiveling. In Li et al., the DNA did not include the upstream −35 region and lacked a complete transcription bubble, which may have prevented observation of overall RNAP conformational changes. In Gao et al., σ[54] has distinct structural features and regulatory mechanisms compared with the mycobacterial σ[E]-RNAP system, likely explaining the absence of swiveling.

Taken together, our data support a model in which *Mtb* RNAP contributes to transcriptional pausing through a series of RNA-induced structural rearrangements triggered by the steric clash between the 5′ end of the nascent RNA and the σ3.2 region. As the RNA extends from 6 to 7 nt, this clash promotes swiveling of the RNAP module and unwinding of the upstream dsDNA, thereby stabilizing the scrunched transcription bubble. These coupled events distort the BH and store free energy, ultimately helping to establish the paused state, consistent with previous biochemical and structural studies that identified a promoter-proximal pause centered on this transition[11–13,64], as well as recent work showing that the 6 to 7 nt transition represents a key branching point leading to either abortive or productive pathways in a sequence-dependent manner[65].

To relate these structural intermediates to de novo initiation, we also determined a complex assembled on a fully matched scaffold in which the 7 nt RNA anneals to positions +1 to +7 of the template strand, matching the register present in our de novo transcription assays (Supplementary Fig. 4, 5 and Supplementary Fig. 1H, I). Although the nucleic-acid density in this matched complex is weaker and does not allow a detailed analysis of the bubble geometry, its overall RNAP conformation is the same as in the pre-melted scaffold, supporting the use of the pre-melted template as a technically necessary and widely adopted strategy for resolving scrunched intermediates in bacterial RNAP complexes (Supplementary Table 1). Consistent with this correspondence, de novo transcription on the pre-melted promoter scaffold also produces prominent 6–7 nt RNA species (Supplementary Fig. 1J, K), indicating that the early-initiation behavior observed structurally is preserved under both matched and pre-melted conditions.

At the same time, we acknowledge that both the use of a pre-melted scaffold and the de novo transcription assays have intrinsic limitations. In early initiation, pausing and abortive release are mechanistically coupled and proceed through the same scrunched intermediate. Even under heparin enforced single-round conditions, bulk assays report only RNA length, and therefore cannot distinguish whether a 6–7 nt RNA arises from a paused or abortive trajectory. This is an intrinsic limitation of bulk de novo transcription and precludes a rigorous kinetic separation of pausing versus abortive release at this stage, rather than a limitation of our specific experimental implementation. The relatively long incubation time used to visualize short RNA species may also allow accumulation of longer products, thereby increasing the apparent run-off/paused ratio without affecting identification of the 6–7 nt intermediate. Furthermore, primer-initiated reactions cannot reproduce the native or pre-melted scaffolds because a fixed RNA primer imposes a register that does not match the 6 to 7 nt configuration captured in our structural analysis, making it unsuitable for probing this transition. For this reason, we used de novo initiation assays, which allow the enzyme to populate the full set of early intermediates without imposing a predefined RNA length. Under such conditions, pausing and abortive release also remain experimentally inseparable, as both derive from the same scrunched intermediate and yield RNA products of identical length[11–13]. The additional RNA band observed between the 6–7 nt species and the run-off product most

likely reflects prematurely terminated or partially extended transcripts generated during early initiation, a common feature of bulk initiation assays, and does not affect the assignment of the dominant 6–7 nt paused population. Nevertheless, the agreement between the matched and pre-melted structures, together with the observed 6–7 nt products in vitro and previous studies that place an initiation pause at the 6 to 7 nt transition[11–13,64,66], supports the view that the paused complex described here represents a physiologically relevant intermediate in σ[E]-dependent transcription initiation. Future single-molecule or rapid-kinetics studies in *Mtb* systems will be important to further dissect the relative contributions of pausing and abortive release at the 6–7 nt checkpoint, building on the structural framework provided here.

## RNA exit channel is the gateway for pausing regulation
The RNA exit channel is formed between the β-flap and β′-clamp subunits of RNAP, where it widens into a positively charged surface that aligns with the phosphates of the A-form RNA, guiding the formation of nascent RNA and serving as a basis for a regulatory connection with the RNAP active site (Fig. 1B, E). Moreover, the RNA exit channel is considered a key regulatory gateway for transcriptional pausing, promoting RNA structure folding or facilitating transcription factor binding, as well as for co-transcriptional processes, mediating interactions with the translational machinery during transcription-translation coupling[67–69]. For example, the formation of PHs and binding of the transcription factor NusA in the RNA exit channel can extend the RNAP dwell time, distinguishing between short and long pauses[47,61,70]. The long pausing state often leads to backtracking, where transcription is blocked until the 3′ end of the nascent RNA is repositioned in the active center either through forward translocation of RNAP or internal RNA cleavage, and can also result in transcription termination, where transcription is halted and the RNA is released[9]. In the structure of PIC, the σ3.2 region threads through the RNAP primary channel, conflicts with the 5′ end of the nascent RNA, and emerges from the RNA exit channel, almost completely occupying it (Fig. 2C–E). Thus, the function of the σ3.2 region is similar to that of PHs in blocking transcription by occupying the RNA exit channel. Our PIC structure, together with previous pause studies, supports a model in which RNAP contributes to transcriptional pausing by occupying or widening the RNA exit channel, promoting swiveling, and blocking transcription[46,59].

## Promoter escape or abortive release
Promoter escape or abortive release remains incompletely understood. Scrunching accumulates stress that can either drive RNA release (abortive initiation) or promoter escape (productive elongation)[28–30]. The formation of the stressed intermediate state provides the free energy needed to release nascent RNA, known as abortive initiation. Initially, RNAP repeatedly synthesizes and releases short RNA transcripts without entering the productive elongation state, as the σ3.2 region occupies the RNA exit pathway (Fig. 5A, B). The release of RNA, rather than productive elongation beyond 9 nucleotides, is a possible mechanism for escaping the stressed intermediate state while being connected to the promoter dsDNA (Fig. 5C, D). Alternatively, accumulated stress provides the driving force for promoter escape, leading to productive elongation (Fig. 5E, F). Thus, the pausing state during transcription initiation likely functions as a regulatory checkpoint that influences promoter escape and abortive release[11–13]. Consistent with these processes, the PIC structure shows that the σ3.2 region is a major determinant in pausing for regulating promoter escape or abortive release. Together with the IC structure, these data suggest that RNAP samples multiple alternative conformational states (IC and PIC) during the transition from 6-nt to 7-nt RNA synthesis within this checkpoint, rather than progressing through a single linear pathway. Additionally, the less-engaged state of the σ4 domain in the PIC structure likely suggests a pathway for σ factor release. Our data suggest that CarD

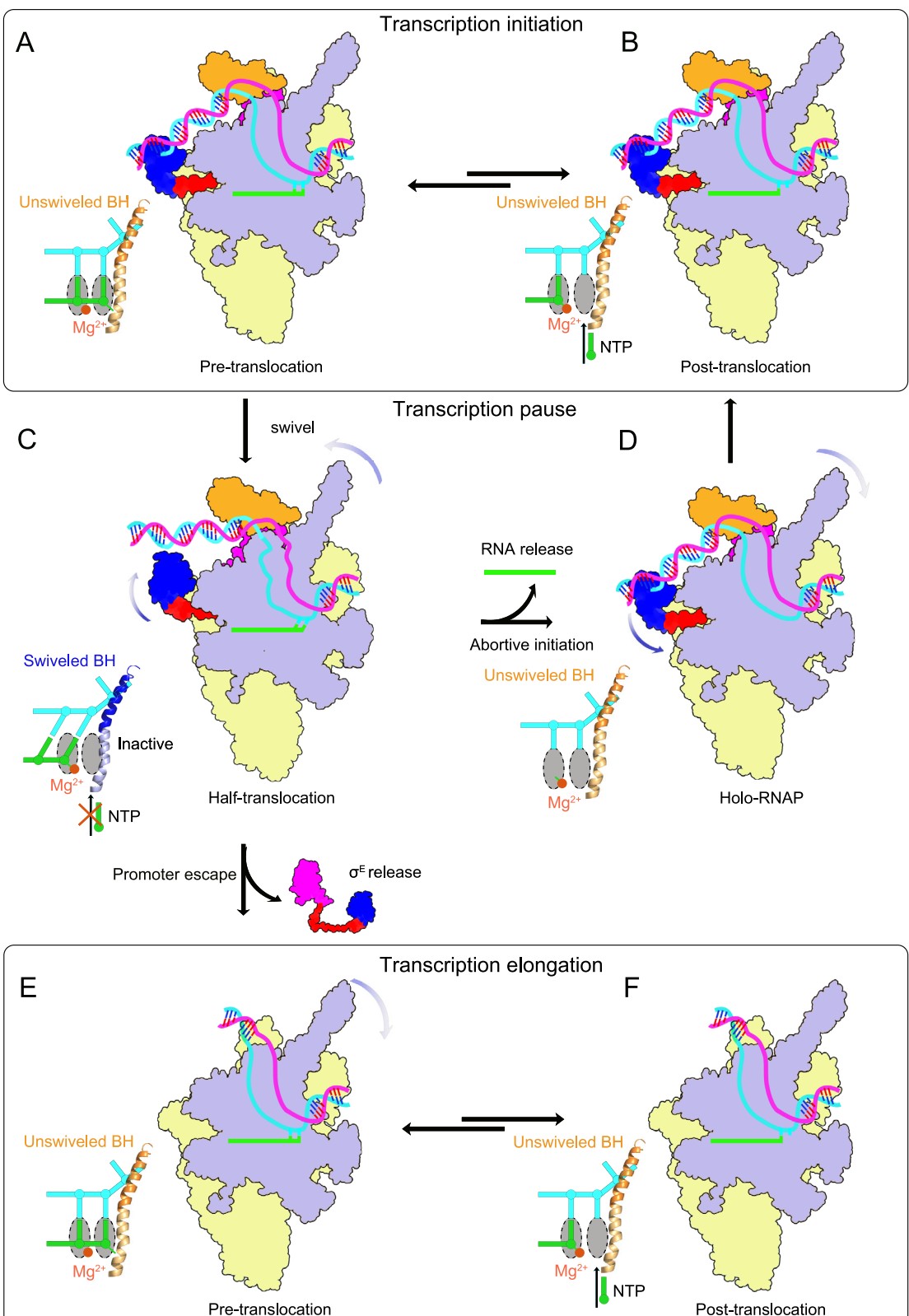

**Fig. 5 | Proposed mechanisms for transcription initiation. A,B** $\sigma^E$ factors facilitate RNAP in forming a transcription bubble and initiating the synthesis of short RNA transcripts in either (**A**) a post-translocation state or (**B**) a pre-translocation state. **C.** Rotation of the swivel module and $\sigma^E$ factors induces a transcriptionally paused state. **D–F** Release of short RNA transcripts leads to abortive initiation (**D**), while dissociation of $\sigma^E$ factors enables promoter escape and transition to an elongation state (**E,F**). RNAP core domain and swivel domain are in yellow and blue, respectively. BH is in blue (PIC) or bright orange (IC or EC). Other molecular colors are consistent with Fig. 1.

may stabilize the transcription bubble during early initiation, helping to prevent premature collapse while maintaining an open configuration that could facilitate the conformational transitions required for escape from the paused intermediate. This potential dual functionality is consistent with its proposed role as both a stabilizer of the RPo and a modulator of productive transcription. Unfortunately, the structures of σ factor release and promoter escape have not yet been determined, leaving the mechanism of how RNAP releases σ factor and escapes the promoter unknown. The PIC structure shows that the σ3.2 region must be displaced by the nascent RNA during transcription elongation, causing the σ4 domain to adopt a reduced engagement with the −35 upstream dsDNA and widening the gap between them. These conformational changes weaken the binding affinity between the σ4 domain and −35 upstream dsDNA, and as RNA elongates, the σ factor is further displaced from the active cleft of RNAP. Therefore, the resolved structure of PIC likely represents a pre-escape state just before the σ factor release and promoter escape. The less-engaged σ4 domain in PIC may indicate a pathway for σ factor release, although the full mechanism remains unresolved. Observed σ[E] retention is consistent with previous findings for σ[70] in *E. coli*[71–73] and σ[A] in *Mtb*[74], supporting the physiological relevance of our initiation and early paused complexes.

## Methods

### Protein production and purification

The *Mtb* RNAP core enzyme was prepared by co-expression of genes for the *Mtb* RNAP α subunit, RNAP ω subunit, RNAP β′ subunit, and C-terminally deca-histidine-tagged RNAP β subunit in *E. coli* [plasmids pRSFDuet-*rpoA*-*rpoZ* and pETDuet-*rpoB*-*rpoC*]. The genomic material was purchased from BeNa Culture Collection. The *Mtb* RNAP core enzyme was expressed in *E. coli* BL21 (DE3) (Supplementary Table 5 and 6). The cells were harvested, resuspended in lysis buffer (20 mM Tris-HCl, pH 8.0, 150 mM NaCl, and 4 mM MgCl$_2$), and homogenized with an ultra-high-pressure cell disrupter at 4 °C. Cell debris was removed by centrifugation at 30,000 x g for 35 min. The *Mtb* RNAP core enzyme was first purified by Ni-NTA (Novagen, USA) affinity chromatography and then further purified by using a Hitrap Q ion-exchange column (GE Healthcare, USA) with buffer A (20 mM Tris-HCl, pH 8.0, 4 mM MgCl$_2$, 2 mM DTT) and buffer B (20 mM Tris-HCl, pH 8.0, 1 M NaCl, 4 mM MgCl$_2$, 2 mM DTT). Next, the sample was loaded onto a Superose6 Increase 10/300 column (GE Healthcare, USA) with DEPC-treated buffer (20 mM HEPES, pH 7.0, 150 mM NaCl, 4 mM MgCl$_2$, and 2 mM DTT). The purified *Mtb* RNAP core enzyme was concentrated to 10 mg/mL and stored at −80 °C (Supplementary Fig. 1A-C).

The *Mtb* σ[E] or its derivative gene was cloned into the pETDuet-1 vector with an N-terminal 6× His-tag. The genomic material was purchased from BeNa Culture Collection, and any mutations were introduced using primers listed in Supplementary Table 6. The protein was expressed in *the E. coli* strain BL21 (DE3) (Supplementary Table 5). Cells were grown at 37 °C and induced at an apparent OD$_{600}$ of 0.6–0.8 by addition of isopropyl-β-D-thiogalactoside (IPTG) to a final concentration of 0.2 mM, and then further grown at 16 °C for 18 h. After harvesting by centrifugation, the cells were resuspended in lysis buffer (20 mM Tris-HCl, pH 8.0, 150 mM NaCl, and 4 mM MgCl$_2$) and homogenized with an ultra-high-pressure cell disrupter at 4 °C. The lysate was centrifuged at 30,000 x g for 35 min to remove cell debris. The fusion protein was first purified by Ni-NTA (Novagen, USA) affinity chromatography and then further purified through passage a Hitrap Q ion-exchange column (GE Healthcare, USA) with buffer A (20 mM Tris-HCl, pH 8.0, 4 mM MgCl$_2$, 2 mM DTT) and buffer B (20 mM Tris-HCl, pH 8.0, 1 M NaCl, 4 mM MgCl$_2$, 2 mM DTT). Next the sample was purified by application to a Superdex 75 10/300 Increase column (GE Healthcare, USA) with DEPC-treated buffer (20 mM HEPES, pH 7.0, 150 mM NaCl, 4 mM MgCl$_2$, 2 mM DTT). Purified *Mtb* σ[E] was concentrated to 10 mg/mL and stored at −80 °C (Supplementary Fig. 1A-C).

*Mtb* CarD or its derivative gene was cloned into the pRSFDuet-1 vector. The genomic material was purchased from BeNa Culture Collection, and any mutations were introduced using primers listed in Supplementary Table 6. CarD with an N-terminal 6× His-tag was expressed in *E. coli* strain BL21 (DE3) (Supplementary Table 5). Cells were grown at 37 °C and induced with 0.2 mM isopropyl-β-D-thiogalactoside at OD$_{600}$ nm 0.6-0.8. After induction at 16 °C for 18 h, the cells were harvested and resuspended in lysis buffer (20 mM Tris-HCl, pH 8.0, 150 mM NaCl, and 4 mM MgCl$_2$) and homogenized with an ultra-high-pressure cell disrupter at 4 °C. After centrifugation at 30,000 x g for 30 min, recombinant protein was purified by Ni-NTA (Novagen, USA) affinity chromatography, and then was further purified by using a Hitrap Q ion-exchange column (GE Healthcare, USA) and a Superdex 75 10/300 Increase column (GE Healthcare, USA) with DEPC-treated buffer (20 mM HEPES, pH 7.0, 150 mM NaCl, 4 mM MgCl$_2$, 2 mM DTT). Purified *Mtb* CarD was concentrated to 10 mg/mL and stored at −80 °C (Supplementary Fig. 1A-C).

### Assembly of the transcription (pausing) initiation complexes

Nucleic acid scaffolds for Cryo-EM complex assembly were prepared from synthetic oligos (Jiangsu GenScript Biotech Co., Ltd, Supplementary Fig. 1D, H and Supplementary Table 7) in DEPC-treated buffer (20 mM HEPES, pH 7.0, 150 mM NaCl, 4 mM MgCl2). Two strategies were used depending on scaffold type: **(i) Pre-melted scaffold**, Template DNA, non-template DNA, and RNA (The 5′ end of the RNA carries a phosphate group) were mixed at a 1:1:1 molar ratio and annealed by heating at 95 °C for 5 min followed by stepwise cooling to 4 °C. The resulting pre-melted scaffold was incubated with *Mtb* RNAP-core at a 1:1.2 molar ratio at 25 °C for 10 min to form a stable RNAP-DNA/RNA complex. The mixture was then incubated with σ[E] and CarD at a 1:3:3 molar ratio to assemble the σ[E]_CarD_RNAP complex. The complexes were purified on a Superose6 Increase 10/300 column (GE Healthcare, USA) equilibrated in the same buffer. Fractions containing *Mtb* RNAP-core, σ[E], and CarD were collected, concentrated to 6 mg/mL, and stored on ice until application to EM grids (Supplementary Fig. 1A-C; scaffold design adapted from ref. 34., with sequences shown in Supplementary Fig. 1D). **(ii) Fully matched scaffold**, Template DNA and RNA (The 5′ end of the RNA carries a phosphate group) were first annealed at a 1:1 molar ratio. The resulting tDNA-RNA duplex was then incubated with *Mtb* RNAP-core at a 1:1.2 molar ratio at 25 °C for 10 min. Subsequently, non-template DNA was added at a 2-fold molar excess relative to the tDNA-RNA duplex and incubated at 25 °C for 30 min, resulting in a stable RNAP-DNA/RNA complex. The mixture was then incubated with σ[E] and CarD at a 1:3:3 molar ratio to assemble the σ[E]_CarD_RNAP complex. The resulting complexes were loaded onto a Superose6 Increase 10/300 column (GE Healthcare, USA) equilibrated in the same buffer. Fractions containing *Mtb* RNAP-core, σ[E], and CarD were collected, identified by SDS-PAGE (Supplementary Fig. 1E, F; gel: Cat No. Gel442012; Biotend, Shanghai, China), concentrated to 6 mg/mL, and stored on ice until application to EM grids (scaffold sequences are shown in Supplementary Fig. 1H and Supplementary Table 7).

### Native electrophoretic mobility shift assays

The binding reaction buffer contained 20 mM HEPES, pH 7.0, 150 mM NaCl, 4 mM MgCl$_2$. A total of 18 μg of *Mtb* RNAP core enzyme was combined with a 1.5 μg nucleic acid scaffold, and σ[E] and CarD were mixed in a 1:2:2 molar ratio. Binding reactions were incubated for 25 min at 30 °C. Samples were run on a 6% polyacrylamide native gel (37.5:1 acrylamide:bis-acrylamide) running in 1×TBE buffer at 100 V for 1.5 h in 4 °C. The gel was stained with ethidium bromide.

The DNA binding affinities of σ[E] was also quantified through EMSA using the same experimental procedure described above, with two

modifications: (1) RNAP-core was omitted from the reaction system, and (2) the molar ratio of DNA to proteins σ$^E$ was adjusted to 1:5.

## Time-resolved fluorescence assay for RPo complex stabilization

The Cy3-labeled template and BHQ-labeled non-template (Supplementary Table 7), with fluorescent labeling performed as previously described[49], were incubated with RNAP core enzyme at 37 °C for 10 min at a 1:2 molar ratio in DEPC-treated buffer containing 20 mM HEPES (pH 7.0), 150 mM NaCl, 4 mM MgCl₂, and 2 mM DTT. σ$^E$ and/or CarD were then added at a molar ratio of 1:5:5 (RNAP-core:σ$^E$:CarD) to assemble the σ$^E$_RNAP_DNA and σ$^E$_CarD_RNAP_DNA complexes. Fluorescence intensity was monitored using a Tecan Spark 20 M multimode microplate reader (Tecan Group Ltd., Switzerland) at 25 °C, with excitation at 510 nm and emission at 570 nm. Data points were collected every 3 s for 900 s. The fluorescence value of DNA-only control ($F_{DNA}$) was subtracted as background, and the fluorescence fold-change was calculated as $(F_t - F_{DNA})/F_{DNA}$. DNA-only, σ$^E$–RNAP–DNA, and σ$^E$–CarD–RNAP–DNA samples were analyzed in parallel. Fluorescence traces were background-corrected, normalized to the initial signal ($F_0$), and fitted to a single-exponential association model, $F_t = F_0 + A(1 - e^{-kt})$, using GraphPad Prism 10 to obtain the apparent rate constant ($k$) and amplitude ($A$). All measurements were performed in triplicate, and data are reported as mean ± SEM from independent experiments.

## Fluorescence-detected in vitro transcription assay

**pCp-AF647-fluorescence-detected.** the σ$^E$-dependent RNAP de novo transcription activity was measured by coupling pCp-AF647-labeled nucleotides to nascent RNA products[75]. Briefly, template DNA was used at a final concentration of 50 nM and incubated with *Mtb* RNAP core enzyme at a final concentration of 200 nM and σ$^E$ at a final concentration of 400 nM in DEPC-treated buffer containing 20 mM HEPES (pH 7.0), 150 mM NaCl, 4 mM MgCl₂, and 2 mM DTT at 37 °C for 15 min to assemble the holo-RNAP complex. CarD was then added at a final concentration of 1 μM, or omitted, and incubated for 5 min to assess its effect on transcriptional pausing. Single-round transcription was ensured by addition of heparin at a final concentration of 50 μg/mL. For reactions using the fully matched DNA template, RNA extension was allowed to proceed for 30 min at 37 °C. For reactions using the pre-melted DNA template, aliquots were collected at 0, 15, 30, and 60 min for analysis. RNA products were extracted with Trizol reagent (VeZol Reagent: Vazyme Biotech Co.,Ltd) and precipitated with isopropanol in the presence of GlycoBlue at a final volume of 1 μl at −20 °C overnight. Precipitated RNA was washed twice with pre-cooled 70% ethanol, air-dried at room temperature, and resuspended in 2.5 μl RNase-free water. The RNA was then ligated with pCp-AF647 using T4 RNA ligase 1. Reactions were stopped by addition of 2× TBE-Urea loading buffer, and RNA-pCp-AF647 products were resolved on freshly prepared 20% urea-polyacrylamide gels containing 8 M urea. Fluorescence signals were detected using a GelView6000 Pro II scanner. All experiments were performed in triplicate.

**Mango-fluorescence-detected.** transcription activity was measured by utilizing the significantly enhanced fluorescence of TO1-3PEG-Biotin when the Mango riboswitch was engaged, as previously reported[76]. Briefly, double-stranded nucleic acid scaffolds for fluorescence-detected in vitro transcription assays were prepared by PCR using pUC57-P*sigB*-*Mango*-tR2 as a template (Supplementary Table 7). The transcription activity of σ$^E$ mutants or CarD mutants was determined in a 20 μL reaction mixture (20 mM HEPES, pH 7.0, 150 mM NaCl, 4 mM MgCl₂, 2 mM DTT) containing the *Mtb* RNAP core enzyme (final concentration 80 nM), promoter DNA or its mutants (final concentration 100 nM), σ$^E$ or its mutants (1 μM), and CarD or its mutants (1 μM) at 37 °C for 10 min. The reaction was initiated by adding 2 μL NTP mixture (ATP, UTP, CTP, and GTP; final

concentration 0.1 mM of each) and 2 μL TOl-3PEG-Biotin (final concentration 0.5 μM), and RNA synthesis was allowed to proceed for 30 min at 37 °C. Fluorescence signals were measured using a Tecan Spark 20 M multimode microplate reader (Tecan Group Ltd., Switzerland) with an excitation wavelength of 510 nm and emission spectra recorded 550 nm. All experiments were performed in triplicate and analyzed using GraphPad Prism 10.

## Cryo-EM sample preparation and data collection

In total, 3.5 μL of protein solution at 6 mg/mL (added with 0.025% DDM, from Merck, supplied by Shanghai YuanXiang Medical Equipment Ltd, China) was applied on a glow-discharged R0.6/1.0 200-mesh Quantifoil grid (Quantifoil, Micro Tools GmbH, Germany). The grid was then blotted for 4.5 s with a blot force of 0 at 8 °C and 100% humidity, and plunge-frozen in liquid ethane using Vitrobot (Thermo Fisher Scientific, USA). The grids were loaded into a 300 keV Titan Krios electron microscope (Thermo Fisher Scientific, USA) equipped with a K3 direct electron detector (Gatan, USA). All images were automatically recorded using a SerialEM with a nominal magnification of 81000× and a calibrated super-resolution pixel size of 0.89 Å/pixel. The exposure time was set to 2.4 s with a total accumulated dose of 55 electrons per Å². The images were collected with a defocus range from −1.0 to −1.8 μm. The statistics for the data collection and refinement are shown in Supplementary Table 2 and 3.

## Cryo-EM image processing

For all samples, the raw movies were motion-corrected and dose-weighted using the MotionCorr2 v1.4.5[77] software, and their contrast transfer functions were estimated using ctffind4 v4.1.3[78]. Particles were automatically picked using a blob picker and extracted with a box size of 320 pixels using cryoSPARC v4.5.1[79]. The following 2D, 3D classifications and refinements were all performed in CryoSPARC. The particles were selected after two rounds of 2D classification based on complex integrity. This particle set was used to perform Ab-Initio reconstruction in eight classes, which were then used as 3D volume templates for heterogeneous refinement. Particle classes containing the complete σ$^E$-CarD-RNAP-DNA/RNA complex were retained for further refinement, while classes lacking one or more components were excluded. As a result, 75,465 particles converged into one *Mtb* IC7 complex class and 65,908 particles converged into one *Mtb* PIC7 complex class. Next, these particles were used for Non-uniform refinement to obtain one class of particles with final resolution of 3.3 Å. The same methods used for IC6 with a 6-nt RNA are described in Supplementary Fig. 2, 3. Similarly, the N-IC6, N-IC7, and N-PIC7 datasets were processed following the same workflow. After 2D and 3D classification, 25,048 particles for N-IC6, 15,872 particles for N-IC7, and 31,912 particles for N-PIC7 were retained for Non-uniform refinement, resulting in final reconstructions at 3.6 Å, 4.0 Å, and 3.9 Å resolution, respectively (Supplementary Fig. 4, 5).

## Model building and refinement

To determine the structure of the *Mtb* IC complex and *Mtb* PIC complex, the structures of the *Mtb* RNAP core enzyme and σ$^H$ complex (PDB: 5ZX2) and CarD (PDB: 6EDT) were individually placed and rigid-body fitted into the cryo-EM map using UCSF Chimera[80]. The DNA/RNA scaffold was manually built in Coot[81] guided by the cryo-EM density and prior scaffold designs (PDB: 6JBQ)[32]. RNA positioning, especially near the σ3.2 region, was adjusted to match the observed density. The complete model was refined in real space using Phenix v 1.19.2[82], with secondary structure and geometry restraints applied. Additional weight optimization was used to balance the fit between protein and nucleic acid components. Model validation was performed using MolProbity for geometry assessment and FSC between model and map for map-to-model fit. The DNA and RNA densities in the active site region were well-resolved, confirming accurate positioning and

correct interactions with protein components. Final refinement statistics are provided in Supplementary Table 2.

## Reporting summary

Further information on research design is available in the Nature Portfolio Reporting Summary linked to this article.

## Data availability

The Cryo-EM density maps and structures have been deposited into the Protein Data Bank (PDB) and Electron Microscopy Data Bank (EMDB) with the accession numbers 9M98 and EMD-63729 for *Mtb* IC6, 9M9D and EMD-63730 for *Mtb* IC7, 9M9E and EMD-63731 for *Mtb* PIC7, EMD-66362 for N-PIC7, EMD-66363 for N-IC6, and EMD-66364 for N-IC7. The PDB entries 8E8M, 8E82, 5UH5, 5ZX2, 6DVD, 6EDT, 6JBQ and 6JCY have been used for structure comparison in this study. Source data are provided with this paper.

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

## Acknowledgements

We thank the staff members at the National Facility for Protein Science in Shanghai Zhangjiang Lab and the cryo-EM centers of CAS Center for Excellence in Molecular Plant Sciences for their technical assistance on cryo-EM data collection. This study was funded by grants from the National Natural Science Foundation of China (32500154 to L.Z.;

32301018 and 32571394 to K.X.), Shanghai Sailing Program (23YF1450100 to L.Z.), Shanghai Health Commission Clinical Research Special Youth Project (20234Y0116 to L.Z.), Tongji University Independent Original Basic Research Project (22120240275 to L.Z.), the National Key Research and Development Program of China (2021YFA1302200 to K.X.), and the Thousand Talents Plan-Youth to K.X., Fundamental Research Funds for the Central Universities (Tongji University) to K.X.

## Author contributions

L.Z. conceived of the project and designed the experiments. L.Z. performed the sample purification and complex assembly. L.Z. collected the cryo-EM data. L.Z. performed Fluorescence-detecting in vitro transcription assays. L.Z. and K.X. analyzed the data. L.Z. and K.X. wrote the manuscript. All authors discussed the experiments, read and approved the manuscript.

## Competing interests

The authors declare no competing interests.
