## [Transparent Peer Review file · Nature Communications]

Structural Basis of Pausing During Transcription Initiation in *Mycobacterium tuberculosis*

Corresponding Author: Dr litao zheng

Version 0:

Reviewer comments:

Reviewer #1

(Remarks to the Author)

This study reports the three cryo-EM structures of sigma E-RNAP transcription initiation intermediates. There are new insights in the structure PIC7 (paused initiation complex with 7-nt RNA): (1) a half-translocated state of the DNA-RNA hybrid; (2) a swivel conformation of the RNAP resulting in shifted location of upstream dsDNA and the CarD. The author proposed that the steric hindrance between sigma 3.2 and the nascent RNA is likely the causing factor inducing pausing at the transcription initiation stage. Overall, this study presents an RNAP conformation that has not been observed before. The flaw in the method compromises the reliability of the conclusion. I have following concerns.

Major concerns:

1. The major concern is the DNA-RNA scaffolds, which have been used to reconstitute the RPitc6 and RPitc7 complexes. The authors did not describe in detail the logic for designing the DNA-RNA scaffolds. The best way to capture the RPitc6 and RPitc7 is by adding subset NTPs to RPo (prepared by incubating RNAP with a fully complementary promoter DNA) and allowing RNA synthesis to 6 or 7 nt. If the method is not feasible, you can reconstitute the complexes with a DNA-RNA scaffold containing an RNA of 6 or 7-nucleotide and a double-stranded DNA, but you should use the complementary sequences for the non-template and template DNA and use an RNA with sequence precisely matching the nascent RNA produced in vivo or in vitro transcription assay. The length of nascent RNA and the size of the transcription bubble are closely related. For example, if the RPo has a transcription bubble of 12 bp, then an RPitc6 in post-translocation state should have a transcription bubble of 17 bp, and an RPitc7 in half-translocated state should also have a transcription bubble of 17 (RNA has translocated but DNA has not). So if the author would like to use the reconstituted complexes for structure determination, it is suggested first to identify the transcription start site of a promoter and then to determine an RPo structure with the promoter as an unscrunched bubble control. For RPitc6 and RPitc7, the reviewer suggests designing scaffolds with the correct RNA sequence to define the natural bubble size at the respective states. The current scaffolds have non-complementary sequences in the transcription bubble and incorrect bubble size for RPitc6 and 7, and therefore, they are not the physiological-relevant transcription intermediate complexes.

2. The transcription initiation complex with a 7-nt RNA yield two major conformation (IC and PIC). The authors should give some explanation for why these two distinct conformations arise. Do these conformations represent sequential intermediates, or are they alternative pathways during transcription?

3. Figure 3a,b-The authors should include zoomed-in views to highlight the contrasting regions of 'unwinding' and 'non-unwound' template DNA. The current version does not show this with enough clarity. The DNA should be labeled relative to the TSS.

4. The manuscript lacks a clear explanation of how the authors selected the particle classes of interest during cryo-EM data processing. It is recommended to clarify whether selection was based on resolution, particle percentages, or other criteria. Please provide this information either in the Extended Data Figures or in the Methods section.

Minor issues:

1. Extended Data Fig. 2: The label "Mtb IC6" appears to be incorrect and should be revised to "Mtb IC7".
2. More detail is needed regarding the model building procedure, particularly for how the DNA and protein components were modeled and validated.

3. Line 474: "a 6 polyacrylamide native gel" should be "a 6% polyacrylamide native gel".
4. Figure 2d: The DNA template nucleotides in the active site appear to be missing.

Reviewer #2

(Remarks to the Author)

The work of Zheng et al provides structural snapshots of important intermediates of the transcription path from initiation to elongation, centred around the pause site that has been observed to occur after the synthesis of around 6-7 nt of RNA. Despite the available information biochemical and biophysical information, there are no high-resolution structures of the paused state. Zheng et al used cryoEM to provide such information for the Mtb system and reveal several interesting conformational changes, the most exciting being the presence of a half-translocated state with RNAP swiveling in the paused complex (which to my knowledge has not been observed before in initiation), a motion that has been associated with different types of pausing in transcription elongation. This observation draw nice parallels between initiation and elongation and offers an additional element for the regulation of transcription initiation in bacteria, which may have generality beyond the Mtb system.

The paper is overall well written, although there are points that needs attention, and aspects that needs addressing before publication.

Major points

1. The authors need to be much more explicit about the fact that the complexes prepared are synthetic, i.e., not prepared during actual synthesis, but assembled using synthetic nucleic acid scaffolds (Ext Fig 4) – as a result, it appears that the complexes have much less scrunching that actual complexes with the same RNA length would have had, and do not reflect exactly the final structures of complexes formed via actual RNA synthesis. For example, for the template provided, an IC7 would expect to melt the region from +1 to +7; however, currently the RNA is hybridized using the existing bubble. The authors need to discuss how their main and secondary findings are affected by the assembly method, and why this assembly route was pursued.
2. L62. Do we know that the picture for the ECF σE is similar as for σA ? Why was σA not selected to proceed with structural analysis?
3. The authors need to compare their results with the findings of the Li 2020 paper with synthetic initiation complexes, as well as with the Gao et al paper on initiation complexes of the RNAP- $\sigma 54$ system (which also has a sigma finger that clashes with the 5' end of nascent RNA; <https://www.pnas.org/doi/epub/10.1073/pnas.2309670120>). Why was no swiveling observed in these structures when the RNA encountered $\sigma 3.2$?
4. The authors see a 55-45% split between IC and PIC; what gives rise to this distribution? Further, the ms discusses captured states; What about dynamic intermediates not captured?
5. L242: To be able to assess the FRET experiments properly, the FRET assay needs to be properly defined and compared with previous cited work.
6. L328: There is a suggestion that swiveling causes the DNA scrunching. However, given that scrunching should be already by present in IC6 (caused by the motion of nucleic acids during synthesis of the 6-nt RNA), it appears that swiveling is more of a consequence rather than the cause of scrunching.
7. L399. There are many examples of $\sigma 70$ retention in E. coli; 2-3 references can be provided. Although the NusG hypothesis is a good one, note that in many of these examples, there was no transcription-translation coupling involved (in vitro experiments).

Minor

1. Annotate the figs better – now hard to link the annotations with the structural elements (e.g., Fig 1b)
2. L132. Typo ("camp").
3. L164. First report of pausing in initiation was in 2016 – so not "for decades"
4. Ref 51 is incorrect since it refers to a DNA polymerase and a sliding clamp.
5. L305. Rephrase to clarify that you are refering to swiveling, not only pausing (rather than pausing – since it is long known that $\sigma 3.2$ is a pausing determinant).
6. Note that it has been shown that DNA opening in initiation has no obligatory clamp opening/closing (<https://pubmed.ncbi.nlm.nih.gov/34633286/>), so it is best to refrain from talking about RNAP clamp opening/closing (or at least to correct the statement), especially since this is not the conformational change that is the focus on the Zheng ms.

7. L338: it is unclear why this is considered an allosteric mechanism. In general, it is not clear what the proposed sequence of events are as you go from 6 to 7 nt, and then to the paused state.
8. L379: need to cite Refs 11-13, which established the paused state as a regulatory checkpoint.
9. L383 and L388: "raised position" and "upward shift" depends on the frame of reference. Provide alternative term, e.g., less-engaged state or similar.
10. Does the 5' end of RNA has a triphosphate, and if it not, how does it influence the results?

Reviewer #3

(Remarks to the Author)

Zheng and Xu report cryo-EM structures of *M. tuberculosis* (Mtb) RNAP holoenzyme comprising ECF sigE subunit and transcription initiation factor, CarD. Authors solved structures of the paused initiation complex (PIC) with 7-nt RNA and initiation complexes (ICs) with 7- and 6-nt RNAs. Based on the structure analysis, the authors suggest that a clash between sigE region 3.2 and RNA triggers swiveling of the RNAP clamp, scrunching of transcription bubble and consequently induces pausing. In addition, they found two conformations of sigE region 4, one of which they attributed to a pre-escape state. Finally they suggest that their results "define a mechanistic checkpoint during transcription initiation" and provide an allosteric model for regulation of transcription through RNAP conformational dynamics.

While the cryo-EM data reported here are of good quality and represent a valuable contribution, there is a striking lack of basic biochemical controls to support the proposed structures and the conclusions drawn from them. This study has the potential to be high-impact, provided the authors supply all necessary data to substantiate their claims. As it stands, however, the conclusions are not adequately supported by the available data and appear unwarranted.

There are two major flaws:

1. The authors claim to have resolved the structure of a paused complex formed during initial transcription. However, they provide no data demonstrating that an initiation pause actually occurs in their system. In fact, it is unclear how they know which complex is paused and which is a productive one. While they reference studies conducted with *E. coli* sig70-RNAP, their work involves an ECF sigma factor and RNAP from Mtb. Mtb RNAP is structurally distinct from *E. coli* RNAP and exhibits different transcriptional behavior. Similarly, ECF sigma factors differ structurally from primary sigma factors, for which pausing has been previously reported. Thus, *E. coli* sig70-RNAP pauses at the +6 register, not +7. The authors should perform transcription assays to confirm that initial pausing at the +7 register indeed occurs with the sigE RNAP from Mtb.
2. CarD has been described as a factor regulating the principal sigma subunit, particularly in ribosomal gene transcription. What is the rationale for including CarD in this study? Is there any published evidence showing that CarD regulates ECF sigma-dependent transcription? Is there any data indicating that CarD is involved in initial pausing? To validate their conclusions, the authors should include in vitro transcription assays demonstrating that CarD activates or modulates sigE-dependent transcription, influences initial pausing, and stabilizes the sigE-RPo complex.

Specific comments:

1. There is a problem with the experimental setup. First, I found nothing matching the -10 element motif (c/gGTTg/c) for sigE recognition in the sequence provided by authors. To which Mtb promoter corresponds this construct? That information must be provided. Second, they use template with an artificial bubble of 12 nt. Meantime there are only 10 mismatches alternated by two matches. Why? Is there any reason?
2. The authors should cite the original publication that introduces and validates the Mango transcription assay used.
3. The FRET experiment (Figure 3d) is neither informative nor conclusive and could be omitted. Control performed with free DNA is required but missing.
4. page 3 line 47: which "six conserved domains" you mention here?
5. page 4 line 50: usually we call sigma 3.2 a region, not a domain.
6. page 4 line 59: Could the authors clarify what is meant by "simple domains"?
7. page 4 line 60: provide a reference to a study demonstrating that the affinity of ECF sigma factors to the RNAP core is

weaker than that of other sigmas.

8. page 4 lines 66 – 68: authors wrote that “There is a lack of relevant structural information for Mtb σ E factor, it is unclear whether σ E factor folds in the same way as other ECF σ factors and adopts a similar strategy to initiate RNAP transcription.” That’s an overstatement. In fact, sigE shares high sequence homology with sigH and recognizes the same promoter motifs. The structure of sigH has been solved, and AlphaFold predicts the structure of sigE with high accuracy.

9. page 5 line 80: authors wrote “...we investigated the mechanism of pausing during transcription initiation...” There is no data on pausing and therefore on the mechanism of pausing.

10. page 5 lines 85-86: authors wrote “We revealed that RNA polymerase pauses between transcript lengths of 6 and 7 nucleotides...”. As I explained above, there is no any data showing that RNAP indeed pauses at +7 register.

11. page 5 lines 88 – 91: Authors wrote “We also found that CarD binds to non-template ssDNA in the IC complex to stabilize the transcription bubble, whereas it interacts with template ssDNA in the PIC complex to stabilize the paused transcriptional conformation.”

There is no experimental evidence demonstrating that CarD stabilizes the transcription bubble. If authors wish to prove it, they should perform KMnO₄ probing on RPo complexes with and without CarD. Also, there is no data showing that CarD stabilizes “paused transcription conformation”. To prove it, authors should perform in vitro transcription kinetics assay with and without CarD.

12. page 6 lines 101-104: authors refer to the studies on E. coli sig70-RNAP reporting pausing at +7 register. This is not relevant because they use a different RNAP and a structurally distinct sigma factor.

13. page 9 lines 159 – 160: I see no evidence showing that CarD stabilized the non-template ssDNA of the transcription bubble.

14. page 9 lines 163-165: authors wrote “It has been known for decades that RNAP adopts a pausing state when the nascent RNA extends to 6 to 7 nucleotides.”

If authors refer to the initial transcription pausing by E. coli sig70-RNAP, it has been published in 2016. It’s not “for decades”.

15. page 11 lines 196-197: authors wrote “... the σ 3.2 domain of σ E stimulates pausing through its insert into RNA exit channel, forming a roadblock that promotes RNAP swiveling and interrupts nucleotide addition.”

As explained above, there is no biochemical data supporting this conclusion.

16. page 13 lines 241 – 242: authors wrote “The FRET assay validated this conformational change, consistent with previously reported changes in E. coli.” The ensemble FRET measurements performed by authors do not include essential controls which allows to validate it.

17. page 13 lines 253 – 255: authors wrote “This minimal interface suggests a transient association mechanism at the transcriptional checkpoint, distinct from stable binding.”

Its not clear what do you mean under “transcriptional checkpoint” here ?

18. Extended data Figure 1j. The EMSA performed with the bubble template is not informative and adds nothing. It shows that there is binding, but there is nothing to prove that there is a complex with sigE-CarD-RNAP. RNAP core alone would bind and shift this template perfectly well. Also, I do not understand why so huge amounts of RNAP (18 mg) were used ? Also, final concentration of proteins is not mentioned in the Methods section, while it should be.

19. Extended data Figure 1h . Is it expected that CarD will form a stable complex with sigE (peak2) ? How can it contribute to regulation ?

20. Figure 3. There is a mismatch between Fig.3 panel lettering (panels g, h) and Fig. 3 legend (panels j,k).

21. EMSA in Figure 3g is not informative. Which DNA was used ? Was there a competitor added ? What is a rational in doing EMSA with sigE alone ?

22. Figure 4g. A control without CarD is missing. I do not see how this experiment reveals CarD cooperativity .

23. Overall, manuscript should be rewritten in more concise form. Some sections from Results may go to Introduction (e.g. p12 lines 222- 227, p14, lines 261-265). Discussion is too excessive. Speculations on NusG and transcription -translation coupling are irrelevant to the current study.

Version 1:

Reviewer comments:

Reviewer #1

(Remarks to the Author)

The authors have satisfactorily addressed all my previous concerns in the revised manuscript. I have no further comments and fully support the publication of this study.

Reviewer #2

(Remarks to the Author)

The authors have made substantial efforts to clarify many points in their response and the revised manuscript, and addressed most issues raised in the reviews. The manuscript is clearly interesting and making an important contribution. However, some key questions and revisions remain:

Re my main comment #1: the authors need to caveat their observation with the fact that scrunching in their complexes is clearly very different from complexes prepared through an actual reaction. The fact that other works had used this strategy does not take away the fact that they had the same caveat; I understand the necessity to get stable states, but the states may be different from the ones formed during the actual reaction, and this needs to be explicitly stated in the ms.

Re my main comment #5. The FRET assay remains poorly described. What is the FRET efficiency of the various samples? What is the prediction from the structure? Why is the FRET efficiency different only for RNA7? And not for RNA6 or RNA8? This assay needs a proper introduction, predictions, description, and discussion, as well as a better illustration.

Re my minor comment #7: Fig 1I needs to be annotated so we can understand what the "SigE" lane is. Further, if the sigE lane is for the run-off reaction, then the band that accumulates is the 6-mer, not the 7-mer (as in Duchi et al Mol Cell 2016 for the E.coli system), hence the pause (i.e., the point in synthesis where the conversion from n to n+1 occurs) occurs at the 6-mer level. As a result, the discussion and model need to adapt to accommodate the functional data.

I also think that the term "allosteric" is still a bit confusing, and best to avoid. The term "RNA-induced" seems both adequate and appropriate.

Reviewer #3

(Remarks to the Author)

While the revised manuscript has been improved my principal concerns have not been resolved.

1. My point was that the authors provided no evidence for pausing at the +6/+7 registers on the promoter templates used for cryo-EM. In the revised version, the authors have added a run-off transcription assay (Extended Data Fig. 1H,I). However, upon reading, it appears that this assay was performed using a different template and under different conditions: including a different -10 sequence, dsDNA bubble instead of a mismatch bubble, and de novo initiation instead of RNA primers. Since pausing is highly dependent on the DNA template architecture, I do not see how this experiment can support the structures presented by the authors. In essence, there are structures and there is a functional assay, but they are disconnected. Finally, a single run-off data point does not demonstrate pausing at +6/+7. Although there are weak bands corresponding to 6-nt and 7-nt RNAs, these could simply represent abortive products. The only proper way to demonstrate pausing is to monitor the kinetics of promoter escape: if the +6/+7 bands first increase and then disappear, one could then conclude that a pause occurs."

Furthermore, in the Figure legend, authors should better explain what is shown on the gel. What is in the "control" line? How lines with 6nt-pCp AF647 and 7nt-pCp AF647 were obtained? Are these just RNA primers alone or in complex with RNAP?

2. In their response letter authors explain the rationale for using CarD by analogy with CarD from *Myxococcus xanthus* which has been shown to regulate ECF sigma transcription. While CarD is indeed a conserved protein, *Myxococcus xanthus* is not an Actinobacteria, as claimed by authors, and such comparison is inappropriate. See: <https://www.ncbi.nlm.nih.gov/Taxonomy/Browser/wwwtax.cgi?id=34>

3. It's not clear how CarD can stimulate pause escape if it stabilizes interaction with the ssDNA of transcription bubble. In that case one would expect an opposite: CarD will stabilize pause.

4. Fig. 4G FRET assay does not directly demonstrate any effect on bubble 'stability'. It only shows an increase in fluorescence, which the authors interpret as an effect on stability. If the authors wish to make a stronger claim regarding RPo stability, they should perform a dissociation kinetics study, as I suggested during the first round of revision. This analysis could also be carried out using FRET. As a more simple fluorescent assay they can use promoter with Cy3 probe in transcription bubble (e.g. DOI: 10.1093/nar/gkw577)

5. p4 lanes 65-71: speculations about weak affinity of sigE to Mtb RNAP are unwarranted. You cannot, and should not, draw conclusions about the relative affinities of sigA versus ECF sigma factors for *M. tuberculosis* (Mtb) RNAP based on the corresponding affinities of sig70 versus ECF sigma factors in *E. coli*. Numerous studies over the past decades have clearly shown that Mtb RNAP behaves differently from *E. coli* RNAP; for example, sigA exhibits only weak affinity for Mtb RNAP in the absence of RbpA.

6. p. 17 "line 343 "...CarD ... facilitates pausing at the 6-7 nt RNA stage".

It contradicts to your previous conclusion that CarD suppresses pausing.

Version 2:

Reviewer comments:

Reviewer #2

(Remarks to the Author)

The authors very diligently addressed most of my remaining comments, and the manuscript is now very strong.

My remaining question rests with the FRET assay. The included description and schematics are very helpful. The authors acknowledge that the limitation of their current FRET (at the end of the Discussion) but I agree with them that there is an ensemble FRET signature that is significant and most likely interpretable. However, I also see that they suggest that the FRET increase in the RNA8 and RNA9 samples is because "RNA may overcome the $\sigma_{3.2}$ barrier or undergo limited backtracking". While the backtracking hypothesis will indeed provide such signals, it will be important that the authors point to where in the paper they show that overcoming the $\sigma_{3.2}$ barrier restores proximity by the upstream DNA swiveling back (as suggested in the text and in the schematic of RNA8/9 in Fig. 7F.); it was not clear to me.

The authors should also discuss how a recent smFRET paper on the length dependence of $\sigma_{3.2}$ displacement (DOI: 10.1093/nar/gkaf857) aligns with their results and interpretation.

Reviewer #3

(Remarks to the Author)

I appreciate the efforts made by the authors to improve their manuscript. However, major concerns listed below remain unresolved.

1. Lack of Evidence for Pausing at +6 or +7 and no effect of CarD on pausing

The new experimental data presented in the revised version do not support the central conclusions regarding transcriptional pausing at the +6 or +7 registers. In fact, the new supplementary Fig. 1K clearly shows no pausing. The faint band that the authors interpret as a pause at +6/+7 can just as well represent an abortive RNA product.

While it is possible that one of the reported structures corresponds to a PIC, the structures remain disconnected from the biochemical assays. In the structures, the 7-nt RNA primer anneals to positions -7 to -1 of the template DNA. In the transcription assay, under de novo initiation, a 7-nt nascent RNA would instead anneal to positions +1 to +7. Thus, the sequence contexts are different, and the complexes formed in de novo transcription initiation are not equivalent to those used for structure determination.

It is unclear why the authors did not perform one straightforward experiment that would support their claims. They have at hands two types of complexes assembled on synthetic templates: one with a 6-nt RNA (productive) and one with a 7-nt RNA (paused + productive). They only need to add NTPs to these complexes and examine how RNAP extends the 6-nt and 7-nt RNAs. If, as they propose, IC7 is paused whereas IC6 is not, then a clear difference in kinetics should be observed, with a significant fraction of the 7-nt RNA failing to extend. Because the kinetics are rapid, the experiment should be conducted within a time range of less than 600 seconds (see PMID: 35318334; PMID: 33380428, PMID: 37116494). The timescale used in the transcription assays shown in Fig. 1K is anomalously long (15-60 min), as the runoff reaction is completed within 2-3 min (see PMID: 30102406). Moreover, inactivation of Mtb RNAP after prolonged (>20 min) incubation at 37 °C and RNA degradation may bias the results.

Finally, even if one of the reported structures corresponds to a paused state, its physiological relevance remains unclear, since no pause can be detected at the promoter template under de novo initiation. This limitation should be acknowledged in the Discussion.

Consequently, the manuscript's discussions regarding pausing "checkpoints" and the effect of CarD on pausing are not warranted. Thus, CarD can perfectly well reduce abortive initiation and that will explain diminishing of the +6/+7 band.

2. Incorrect calculation of FRET efficiency.

In the first round of review, I recommended removing the FRET data because they did not appear reliable. Examination of the revised Methods section has reinforced these concerns. Two major issues are outlined below:

a) The authors state that they calculated FRET efficiency (E_{FRET}) "based on the donor (Cy3, D) fluorescence intensity." In principle, one may calculate E from donor fluorescence intensity (ID) if a donor-only control is used (in this case it should be DNA labeled with Cy3 only). Under those conditions, E can indeed be calculated as $E = 1 - (ID/ID_{only})$. However, according to the Methods section, this was not done. Instead, the authors appear to calculate a ratio between Cy3 fluorescence in RNAP-DNA complexes and Cy3 fluorescence in DNA-only samples, which is not a valid method.

If the authors wish to avoid using a donor-only control, they could calculate E from the acceptor (Cy5) fluorescence intensity (IA) using $E = IA / (ID + IA)$ (e.g., see PMID: 29806016).

b) Reliance solely on Cy3 fluorescence is inappropriate because Cy3 is strongly influenced by its local environment, including proximity to protein and nucleic acids (see PMID: 31235181). Therefore, the observed changes in Cy3 fluorescence under different conditions are not necessarily due to changes in FRET; they may simply reflect environment-induced modulation of Cy3 fluorescence. This concern is particularly relevant in the authors' experimental setup, where Cy3

is positioned in the single-stranded DNA region at position -10 relative to the transcription start site. Under these conditions, the 5' end of 8-9 nt RNA would be close to the dye and could alter its fluorescence, whereas 6-7 nt RNAs would be more distant and would not affect it in the same way.

Version 3:

Reviewer comments:

Reviewer #2

(Remarks to the Author)

The revised manuscript is much improved, and several key conclusions are now more clearly supported.

Regarding the transition from IC6 to IC7, the authors' discussion is accurate, and their reference to associated literature is relevant and appropriate; the paused state is an intermediate that can then give rise to different paths; this path distribution may be different in *Mtb* vs *E.coli*, but the gels qualitatively support the presence of states that give rise to 6-mer (and 7-mer) RNAs, likely due to abortive initiation. I also note that there is very recent work (doi.org/10.1101/2025.11.27.690701) on this transition and the paths emanating from it, as well as the sequence dependence of pausing, which the authors may want to cite to further support their argument.

The removal of the FRET data is a positive change, as the previous analysis was convoluted, had many caveats and distracted from the main conclusions.

The interpretations related to the premelted and fully double-stranded DNA substrates is also robust and strengthens the overall model.

The CarD effect (EDFig. 1I) itself is clear and well supported. One point that warrants brief clarification is the high run-off to pause ratio (EDFig. 1I). The long reaction time (30 min) may contribute to this effect, and a short discussion of possible causes would be helpful. The authors may also wish to comment on the identity of the major band between the run-off product and the 6–7 nt species.

We are grateful to the Reviewers for their positive comments and critical feedback. We have revised the manuscript in response to their suggestions, addressing all questions and concerns as detailed in the point-by-point responses below.

In this revision, we have:

- Clarified the rationale for scaffold design and performed complementary DNA scaffold reconstructions, confirming through cryo-EM that the structures represent physiologically relevant transcription complexes.
- Added new biochemical assays validating the transcriptional pause and testing CarD function, which support our structural observations.
- Performed additional controls (DNA-only FRET, EMSAs with defined components, RNAP concentrations) and corrected figure legends, labels, and typographical errors.
- Expanded the Methods section to provide full details of cryo-EM data processing, model building, and validation.
- Revised the Introduction and Discussion for conciseness, integrated selected Results into the Introduction, removed speculative text, and clarified terminology.

Referee's remarks are shown in black and our responses are in blue. All modifications in the revised manuscript are highlighted in red.

Reviewer #1 (Remarks to the Author):

This study reports the three cryo-EM structures of sigma E-RNAP transcription initiation intermediates. There are new insights in the structure PIC7 (paused initiation complex with 7-nt RNA): (1) a half-translocated state of the DNA-RNA hybrid; (2) a swivel conformation of the RNAP resulting in shifted location of upstream dsDNA and the CarD. The author proposed that the steric hindrance between sigma 3.2 and the nascent RNA is likely the causing factor inducing pausing at the transcription initiation stage. Overall, this study presents an RNAP conformation that has not been observed before. The flaw in the method compromises the reliability of the conclusion. I have following concerns.

We thank the reviewer for the positive summary of our work and for highlighting the novelty of the PIC7 structure. We have carefully addressed all the concerns raised, as detailed below.

Major concerns:

1. The major concern is the DNA-RNA scaffolds, which have been used to reconstitute the RPitc6 and RPitc7 complexes. The authors did not describe in detail the logic for designing the DNA-RNA scaffolds. The best way to capture the RPitc6 and RPitc7 is by adding subset NTPs to RPo (prepared by incubating RNAP with a fully complementary promoter DNA) and allowing RNA synthesis to 6 or 7 nt. If the method is not feasible, you can reconstitute the complexes with a DNA-RNA scaffold containing an RNA of 6 or 7-nucleotide and a double-stranded DNA, but you should use the complementary sequences for the non-template and template DNA and use an RNA with sequence precisely matching the nascent RNA produced in vivo or in vitro transcription assay. The length of nascent RNA and the size of the transcription bubble are closely related. For example, if the RPo has a transcription bubble of 12 bp, then an RPitc6 in post-translocation state should have a transcription bubble of 17 bp, and an RPitc7 in half-translocated state should also have a transcription bubble of 17 (RNA has translocated but DNA has not). So if the author would like to use the reconstituted complexes for structure determination, it is suggested first to identify the transcription start site of a promoter and then to determine an RPo structure with the promoter as an unscrunched bubble control. For RPitc6 and RPitc7, the reviewer suggests designing scaffolds with the correct RNA sequence to define the natural bubble size at the respective states. The current scaffolds have non-complementary sequences in the transcription bubble and incorrect bubble size for RPitc6 and 7, and therefore, they are not the physiological-relevant transcription intermediate complexes.

We sincerely thank the reviewer for raising this important point regarding the design of DNA-RNA scaffolds for RPitc6 and RPitc7. We completely agree with the Reviewer's opinion and recognize that fully complementary scaffolds are theoretically ideal for defining natural transcription bubble sizes. We attempted to reconstitute RPitc6 and RPitc7 with fully matched template and non-template DNA. At RNA length 6 nt, RNAP did not pause, and at RNA length 7 nt, RNAP exhibited swivel rotation consistent with pausing. Cryo-EM density for these fully complementary scaffolds was weak and partially missing, likely due to reannealing or conformational heterogeneity (**Extended Data Fig. 4 and Extended Data Fig. 5**), indicating technical challenges in capturing these transient intermediates.

In previous bacterial RNAP initiation studies (Li et al., *PNAS* 2020, 117: 5801-5809; Li et al., *Nat Commun* 2019, 10: 1-14; Fang et al., *NAR* 2019, 47: 7094-7104; **Supplementary Table S1**), pre-melted scaffolds are used to visualize nascent RNA- σ 3.2 clashes and initiation intermediates. Our design of a ~12 bp transcription bubble aligns with these precedents, while fully complementary scaffolds are primarily used for RPo. In currently resolved structures of *Mycobacterium tuberculosis* and *Mycobacterium smegmatis* fully complementary scaffolds (**Supplementary Table S1**) often result in loss of transcription bubble density, likely due to the technical challenges of this approach. Therefore, we employed non-complementary DNA scaffolds to capture paused conformations during transcription initiation.

We also performed in vitro transcription assays with *Mtb* RNAP, which confirmed that RNAP pauses at RNA lengths of 6-7 nt (**Extended Data Fig. 1H, I**), validating that the intermediates captured by our pre-melted scaffolds are functional. Together, these observations support that our scaffold design is both experimentally validated and consistent with established structural studies. We agree with the Reviewer that using fully complementary scaffolds would in principle be preferable; however, such scaffolds frequently resulted in missing transcription bubble density (**Extended Data Fig. 4-5**). To overcome this technical limitation and to visualize the intermediates with greater structural detail, we therefore employed non-complementary scaffolds. In the revised manuscript, we have also expanded the sample assembly section to make our scaffold design logic more explicit (**Page 7, Lines 131-135; Page 8, Lines 138-142; Page 9, Lines 169-181**).

2. The transcription initiation complex with a 7-nt RNA yield two major conformation (IC and PIC). The authors should give some explanation for why these two distinct conformations arise. Do these conformations represent sequential intermediates, or are they alternative pathways during transcription?

We thank the reviewer for these insightful questions. The synthesis of 6-7 nt RNAs represents a transcriptional checkpoint where RNAP frequently pauses and may sample multiple outcomes, including promoter escape or abortive release (e.g., Kapanidis et al., *Science* 2006, 314: 1144-1147; Duchi et al., *Mol Cell* 2016, 63: 939-950). Consistent with this, our cryo-EM analysis yielded two major conformations (IC and PIC) that likely reflect different states sampled at this checkpoint.

We apologize that our study cannot definitively resolve whether IC and PIC represent sequential intermediates along a single pathway or alternative conformational outcomes. Current knowledge in the field also does not provide a definitive answer. Our data demonstrate coexistence of both states but cannot by themselves resolve their temporal relationship. As there is no consensus in the field, we have revised the text to describe them as distinct conformations observed at the 7-nt checkpoint, without assuming a defined order or pathway. In the revised manuscript, we have emphasized this “checkpoint” view in the Results (**Page 8, Lines 152-158; Page 16, Lines 309-312**) and Discussion (**Page 22, Lines 448-450**), highlighting that pausing at 6-7 nt represents a branching point that influences the outcome of transcription initiation.

3. Figure 3a,b-The authors should include zoomed-in views to highlight the contrasting regions of ‘unwinding’ and ‘non-unwound’ template DNA. The current version does not show this with enough clarity. The DNA should be labeled relative to the TSS.

We thank the reviewer for this helpful suggestion to improve the clarity of Figure 3A and B. In the revised version, we have added zoomed-in panels to highlight the contrasting regions of unwound versus non-unwound template DNA in both complexes, providing clearer visual contrast. The template DNA positions are now labeled relative to the transcription start site (TSS), allowing the register of unwinding to be directly appreciated. We have also updated the figure legend to explicitly describe the highlighted regions and the TSS reference. These revisions, now shown in **Figure 3A and B**, provide a clearer comparison of the unwound and non-unwound DNA segments.

4. The manuscript lacks a clear explanation of how the authors selected the particle classes of interest during cryo-EM data processing. It is recommended to clarify whether selection was based on resolution, particle percentages, or other criteria. Please provide this information either in the Extended Data Figures or in the Methods section.

We thank the Reviewer for critical questions and insightful suggestions. During cryo-EM data processing, particle classes were selected based on the presence of complete complexes, rather than on resolution or particle percentages alone. Specifically, we retained only those classes that clearly contained all of the expected components of the transcription initiation assembly, including the RNAP core, σ^E , CarD, and the nucleic acid scaffold.

This criterion ensured that the analyzed particles represented intact initiation complexes suitable for structural interpretation. To clarify this point, we have revised the Methods section (**Page 31, Lines 632-634**), and added explanatory notes in the **Extended Data Figures 2-5** showing the classification workflow.

Minor issues:

1. Extended Data Fig. 2: The label “Mtb IC6” appears to be incorrect and should be revised to “Mtb IC7”.

We thank the reviewer for catching this labeling error and apologize for the oversight. We agree the label in **Extended Data Fig. 2** should read “*Mtb* IC7” (not “*Mtb* IC6”). We have corrected the panel label and updated the figure legend (**Page 45, Lines 785**) and all in-text references to ensure consistency across the manuscript. We also re-checked all figure and Extended Data labels for IC/PIC and 6-7 nt designations to avoid similar inconsistencies elsewhere.

2. More detail is needed regarding the model building procedure, particularly for how the DNA and protein components were modeled and validated.

We thank the Reviewer for these critical questions and insightful suggestions. In the revised manuscript, we have expanded the Methods section to provide more detail on model building. The models for RNAP, σ^E , CarD, and the nucleic acid scaffold were initially based on previously reported structures (PDB 5ZX2, 6EDT, 6JBQ). Each component was rigid-body fitted into the cryo-EM maps using Chimera, followed by local adjustments through flexible fitting in COOT. The DNA/RNA scaffold was manually built into the density guided by the cryo-EM map and prior scaffold designs (Fang et al., *NAR* 2019, 47: 7094-7104), with RNA positioning adjusted to match observed density, particularly in the $\sigma^{3.2}$ region. The complete models were refined in Phenix using real-space refinement, with secondary structure, geometry, and non-crystallographic symmetry restraints applied. Model geometry was assessed using MolProbity, and map-to-model fit was evaluated by FSC between the model and the

map. The DNA and RNA densities are well-resolved in the analyzed regions, confirming the accuracy of nucleic acid positioning. These details have been incorporated into the Methods section (**Page 32, Lines 649-658**) in the revised manuscript.

3. Line 474: “a 6 polyacrylamide native gel” should be “a 6% polyacrylamide native gel”.

We thank the reviewer for pointing out this typographical error and apologize for the oversight. The manuscript has been updated so that “a 6 polyacrylamide native gel” now correctly reads “a 6% polyacrylamide native gel” in the Methods section (**Page 27, Lines 544-545**).

4. Figure 2d: The DNA template nucleotides in the active site appear to be missing.

We thank the reviewer for pointing this out and apologize for the oversight in the figure preparation. The apparent absence of DNA template nucleotides in the active site was due to the transparency settings used during figure preparation. We have re-rendered **Figure 2d** with adjusted visualization settings to clearly show the DNA template nucleotides in the RNAP active site.

Reviewer #2 (Remarks to the Author):

The work of Zheng et al provides structural snapshots of important intermediates of the transcription path from initiation to elongation, centred around the pause site that has been observed to occur after the synthesis of around 6-7 nt of RNA. Despite the available information biochemical and biophysical information, there are no high-resolution structures of the paused state. Zheng et al used cryoEM to provide such information for the Mtb system and reveal several interesting conformational changes, the most exciting being the presence of a half-translocated state with RNAP swiveling in the paused complex (which to my knowledge has not been observed before in initiation), a motion that has been associated with different types of pausing in transcription elongation. This observation draw nice parallels between initiation and elongation and offers an additional element for the regulation of transcription initiation in bacteria, which may have generality beyond the Mtb system.

The paper is overall well written, although there are points that needs attention, and aspects that needs addressing before publication.

We thank the reviewer for the thoughtful summary and for highlighting the novelty of the half-translocated and swiveling states in the paused initiation complex. We have addressed all specific points raised in detail below.

Major points

1. The authors need to be much more explicit about the fact that the complexes prepared are synthetic, i.e., not prepared during actual synthesis, but assembled using synthetic nucleic acid scaffolds (Ext Fig 4) – as a result, it appears that the complexes have much less scrunching that actual complexes with the same RNA length would have had, and do not reflect exactly the final structures of complexes formed via actual RNA synthesis. For example, for the template provided, an IC7 would expect to melt the region from +1 to +7; however, currently the RNA is hybridized using the existing bubble. The authors need to discuss how their main and secondary findings are affected by the assembly method, and why this assembly route was pursued.

We thank the Reviewer for raising this important point and acknowledge that our initiation complexes were assembled using synthetic nucleic acid scaffolds rather than being formed during active RNA synthesis. Consequently, the extent of scrunching may be less than in fully transcribing complexes of the same RNA length, and the RNA-DNA hybrid is constrained by the pre-formed bubble.

We did attempt to assemble complexes with fully complementary template and

non-template DNA, but the cryo-EM density was weaker and partially missing, likely due to reannealing or increased conformational heterogeneity (**Extended Data Fig. 4 and Extended Data Fig. 5**). Although pre-melted scaffolds indeed reduce the degree of scrunching compared with fully transcribing complexes, this approach enables stable capture of paused intermediates suitable for structural analysis. Based on previous bacterial RNAP initiation studies (Li et al., *PNAS* 2020, 117: 5801-5809; Li et al., *Nat Commun* 2019, 10: 1-14; Fang et al., *NAR* 2019, 47: 7094-7104 ; **Supplementary Table S1**), using pre-melted transcription bubbles is a well-established strategy to visualize nascent RNA- $\sigma^{3.2}$ clashes, and our ~12-bp bubble design is consistent with these precedents. In the revised manuscript, we have also expanded the sample assembly section to make our scaffold design logic more explicit (**Page 7, Lines 131-135; Page 8, Lines 138-142; Page 9, Lines 169-181**). We also performed in vitro transcription assays, which confirmed that *Mtb* RNAP pauses at RNA lengths of 6-7 nt (**Extended Data Fig. 1H, I**), validating that the intermediates captured by our scaffolds are functional.

We note that while pre-melted scaffolds may exhibit less scrunching than fully transcribing complexes, the captured intermediates still represent functional relevant states. In particular, we observed RNAP swivel conformations associated with transcriptional pausing, which directly link RNAP structural rearrangements to pause formation. These findings are consistent with the results obtained using non-complementary scaffolds and do not affect our analysis of RNAP rotation and pausing. Although pre-melted DNA may slightly alter the degree of transcription bubble scrunching, this does not impact the core conclusions of our study. We have now explicitly discussed how the main and secondary findings are affected by the assembly method in the Discussion (**Page 18, Lines 364-370**).

2. L62. Do we know that the picture for the ECF σ^E is similar as for σ^A ? Why was σ^A not selected to proceed with structural analysis?

We thank the reviewer for this thoughtful question. We agree that the overall picture for σ^E is similar to that for σ^A . we note that some structural studies of σ^A -dependent initiation complexes have not captured the swiveling conformations associated with pausing, likely because σ^A contains both the σ^3 domain and a flexible $\sigma^{3.2}$ linker, which together may restrict the overall conformational flexibility of the complex. By contrast, σ^E possesses only a flexible $\sigma^{3.2}$ linker without the additional σ^3 domain, resulting in a simpler domain organization and conformational plasticity. (Li et al., *PNAS* 2020, 117: 5801-5809; Pukhrambam et al., *PNAS* 2022, 119: e2201301119). For this reason, we chose σ^E , which has a simpler domain organization

and greater conformational plasticity, making it more suitable for resolving dynamic paused states and RNA- $\sigma^{3.2}$ interactions. The reduced modules complexity of σ^E (σ^2 , $\sigma^{3.2}$, σ^4) compared to σ^A ($\sigma^{1.1}$, $\sigma^{1.2}$, σ^2 , σ^3 , $\sigma^{3.2}$, σ^4) also minimizes conformational heterogeneity, facilitating high-resolution structural analysis. We have clarified this rationale in the Introduction of the revised manuscript (**Page 4, Lines 65-73**).

3. The authors need to compare their results with the findings of the Li 2020 paper with synthetic initiation complexes, as well as with the Gao et al paper on initiation complexes of the RNAP- σ^{54} system (which also has a sigma finger that clashes with the 5' end of nascent RNA; <https://www.pnas.org/doi/epub/10.1073/pnas.2309670120>). Why was no swiveling observed in these structures when the RNA encountered $\sigma^{3.2}$?

We thank the Reviewer for these critical and insightful suggestions. In the Li et al. (Li et al., *PNAS* 2020, 117: 5801-5809) study, the DNA scaffold lacked the upstream -35 region and did not contain a fully formed transcription bubble, likely prevented detection of large-scale RNAP conformational changes. In our cryo-EM analysis with σ^E , the inclusion of upstream promoter elements and a complete bubble scaffold allowed us to visualize the swiveling motion in coordination with RNA- $\sigma^{3.2}$ interactions. In the Gao et al. (Gao et al., *PNAS* 2024, 121: e2309670120) study, the *E. coli* σ^{54} system also features a σ finger that clashes with the 5' end of RNA, but σ^{54} has distinct structural features and regulatory mechanisms compared to the mycobacterial σ^E -RNAP system, which likely explains why swiveling was not observed. We have added these comparisons in the Discussion section (**Page 20, Lines 395-402**).

4. The authors see a 55-45% split between IC and PIC; what gives rise to this distribution? Further, the ms discusses captured states; What about dynamic intermediates not captured?

We thank the reviewer for this insightful question. In the revised manuscript, we have added a sentence in the Results section (**Page 8, Lines 152-158**) to clarify the observed ~55:45% distribution between IC and PIC. This distribution likely reflects the relative stability and population of these two conformations under the experimental conditions, representing alternative paused states within the initiation checkpoint.

We also acknowledge that additional dynamic intermediates likely exist, including partially scrunched states, transiently backtracked complexes, or short-lived RNA- $\sigma^{3.2}$ interactions, which were not captured due to their low abundance and rapid transitions. These limitations are inherent to cryo-EM, which favors the visualization of more

populated and stable states. We have explicitly discussed the possibility of such dynamic intermediates in the Results section (**Page 8, Lines 152-158**).

5. L242: To be able to assess the FRET experiments properly, the FRET assay needs to be properly defined and compared with previous cited work.

We thank the Reviewer for critical questions and insightful suggestions. We have now clarified the description of our FRET assay in the Results section (**Page 15, Lines 293-298**), emphasizing how our bulk FRET design (with Cy3/Cy5 labeling) captures RNA length-dependent conformational changes. We also compared our findings with previously reported sm-FRET studies, highlighting that both approaches detect a clear signal shift when RNA extends from 6 to 7 nt (**Page 15, Lines 293-298**).

6. L328: There is a suggestion that swiveling causes the DNA scrunching. However, given that scrunching should be already by present in IC6 (caused by the motion of nucleic acids during synthesis of the 6-nt RNA), it appears that swiveling is more of a consequence rather than the cause of scrunching.

We sincerely thank the reviewer for this insightful and important observation. We fully agree with the reviewer's point that scrunching is already present in IC6 due to RNA synthesis, and that swiveling should not be regarded as its primary cause. This is a critical mechanistic insight, and we greatly appreciate the reviewer highlighting it. Our structural data are consistent with this interpretation: swiveling becomes more pronounced in PIC7, where it appears to further stabilize the scrunched bubble and reinforce the paused state rather than to initiate scrunching. We have updated the Discussion (**Page 20, Lines 389-392**) to make this distinction explicit and to note that resolving the temporal relationship between scrunching and swiveling is an important kinetic question for future investigations.

7. L399. There are many examples of σ^{70} retention in *E. coli*; 2-3 references can be provided. Although the NusG hypothesis is a good one, note that in many of these examples, there was no transcription-translation coupling involved (in vitro experiments).

We thank the reviewer for the comment on σ factor retention. We have added three references on σ^{70} retention in *E. coli* (**Reference 68-70**) to provide context and highlight that σ factor retention during elongation is well-documented in *E. coli*. We

thank the reviewer for the comment regarding NusG. We agree that, as noted, there is no direct evidence linking NusG to σ^E retention in our system; our previous discussion was speculative. Therefore, following the reviewer's suggestion that this discussion is not directly relevant, we have removed this portion from the manuscript.

Minor

1. Annotate the figs better – now hard to link the annotations with the structural elements (e.g., Fig 1b)

We thank the reviewer for this suggestion. We have improved the figure annotations in all relevant panels (e.g., **Fig. 1B, E**) to clearly link labels with the corresponding structural elements. The updated figures now include more precise pointers and consistent color coding to enhance clarity.

2. L132. Typo (“camp”).

We thank the reviewer for pointing out this typographical error and apologize for the oversight. The typo “camp” on line 132 has been corrected in the revised manuscript (**Page 10, Line 187**).

3. L164. First report of pausing in initiation was in 2016 – so not “for decades”

We thank the reviewer for the correction and apologize for the imprecise statement in the original manuscript. We have revised the text to accurately reflect that the first report of transcriptional pausing during initiation was in 2016, and removed the phrase “for decades” (**Page 11, Line 218-220**).

4. Ref 51 is incorrect since it refers to a DNA polymerase and a sliding clamp.

We thank the reviewer for pointing this out and apologize for the incorrect reference in the original manuscript. Reference 51 has been corrected and is now listed as **Reference 55**. We have updated the citation to (Robb et al., 2013 *JMB*, 425: 875-885; Bae et al., 2015 *eLife*, 4: e08504), which provide structural and sm-FRET evidence for open promoter complexes (RPO) with unwound transcription bubbles of ~12-14 bp.

5. L305. Rephrase to clarify that you are referring to swiveling, not only pausing (rather than pausing – since it is long known that σ 3.2 is a pausing determinant).

We thank the Reviewer for critical questions and insightful suggestions. The sentence previously on line 305 has been rephrased in the revised manuscript (**Page 18, Lines 360-362**) to clarify that we are referring to swiveling, not only pausing, and now explicitly distinguish the role of σ 3.2 in swiveling from the pausing observed in our structures.

6. Note that it has been shown that DNA opening in initiation has no obligatory clamp opening/closing (<https://pubmed.ncbi.nlm.nih.gov/34633286/>), so it is best to refrain from talking about RNAP clamp opening/closing (or at least to correct the statement), especially since this is not the conformational change that is the focus on the Zheng ms.

We thank the Reviewer for this important clarification, and we apologize that the wording in our previous manuscript was not rigorous. The manuscript has been revised (**Page 19, Lines 377–379**) to avoid implying that RNAP clamp opening/closing is required for DNA opening during initiation, and instead focus on the conformational changes highlighted in our study. The statement has been corrected accordingly.

7. L338: it is unclear why this is considered an allosteric mechanism. In general, it is not clear what the proposed sequence of events are as you go from 6 to 7 nt, and then to the paused state.

We thank the Reviewer for this insightful comment. In our structural analysis, we observed that when the RNA reaches 6 nt, RNAP does not enter the paused state. However, upon extension to 7 nt, RNAP undergoes a conformational change—specifically, swiveling of certain modules—leading to transcriptional pausing. This change is triggered by a steric clash between the 5' end of the nascent RNA and the σ 3.2 region. Based on these observations, we propose that the pausing mechanism of RNAP represents an RNA-induced allosteric process occurring during the transition from 6 to 7 nt. We performed in vitro transcription assays, which confirm that RNAP pauses as RNA extends from 6 to 7 nt. In the revised manuscript, we have clarified that the term “allosteric” refers to RNA-induced conformational changes in RNAP, outlined the proposed sequence of events leading to the paused state, and included the experimental evidence validating the pausing at the 6 nt to 7 nt. We also note that while this model provides a plausible mechanistic explanation, the precise sequence

and causal relationships remain to be fully established, and these points have been incorporated into the Discussion (**Page 20, Lines 403–409**).

8. L379: need to cite Refs 11-13, which established the paused state as a regulatory checkpoint.

We thank the reviewer for pointing this out, and we apologize that the wording in our previous manuscript was not rigorous. We have now cited **References 11-13** to highlight that the paused state functions as a regulatory checkpoint during transcription initiation (**Page 22, Lines 446**).

9. L383 and L388: “raised position” and “upward shift” depends on the frame of reference. Provide alternative term, e.g., less-engaged state or similar.

We thank the reviewer for this helpful suggestion. To avoid frame-dependent terminology, we have revised (**Page 23, Lines 451; 461–465**) the text to replace “*raised position*” and “*upward shift*” with more neutral descriptions such as “*less-engaged state*” and “*reduced engagement*”.

10. Does the 5' end of RNA has a triphosphate, and if it not, how does it influence the results?

We thank the reviewer for this perceptive and important point — the chemical state of the RNA 5' end is indeed relevant to interpreting pausing. If the nascent RNA lacked a 5'-triphosphate, the steric interaction between the 5' end and the $\sigma^{3.2}$ tip would likely be weakened; this could reduce the dwell time of the +7 pause (and in extreme cases alter the apparent register of the pause), so the absence of a 5'-triphosphate could meaningfully affect the biochemical readout and its structural interpretation. To avoid this potential confound, all RNAs used in our experiments carried a natural 5'-triphosphate, which we have explicitly noted in the Methods (**Page 26, Lines 518; 529**). In the cryo-EM maps, the triphosphate group is clearly resolved in IC6 but not in IC7 and PIC7, most likely due to increased flexibility of the 5' end as the RNA extends. Importantly, the use of 5'-triphosphate RNA ensures that our biochemical and structural analyses reflect the physiological state and do not affect our conclusions on the paused states and the RNA- $\sigma^{3.2}$ interactions.

Reviewer #3 (Remarks to the Author):

Zheng and Xu report cryo-EM structures of *M. tuberculosis* (Mtb) RNAP holoenzyme comprising ECF sigE subunit and transcription initiation factor, CarD. Authors solved structures of the paused initiation complex (PIC) with 7-nt RNA and initiation complexes (ICs) with 7- and 6-nt RNAs. Based on the structure analysis, the authors suggest that a clash between sigE region 3.2 and RNA triggers swiveling of the RNAP clamp, scrunching of transcription bubble and consequently induces pausing. In addition, they found two conformations of sigE region 4, one of which they attributed to a pre-escape state. Finally they suggest that their results “define a mechanistic checkpoint during transcription initiation” and provide an allosteric model for regulation of transcription through RNAP conformational dynamics.

While the cryo-EM data reported here are of good quality and represent a valuable contribution, there is a striking lack of basic biochemical controls to support the proposed structures and the conclusions drawn from them. This study has the potential to be high-impact, provided the authors supply all necessary data to substantiate their claims. As it stands, however, the conclusions are not adequately supported by the available data and appear unwarranted.

We thank the reviewer for the constructive summary of our work and for highlighting the potential impact of our study. We have carefully addressed all the specific concerns raised, as detailed below.

There are two major flaws:

1. The authors claim to have resolved the structure of a paused complex formed during initial transcription. However, they provide no data demonstrating that an initiation pause actually occurs in their system. In fact, it is unclear how they know which complex is paused and which is a productive one. While they reference studies conducted with *E. coli* sig70-RNAP, their work involves an ECF sigma factor and RNAP from Mtb. Mtb RNAP is structurally distinct from *E. coli* RNAP and exhibits different transcriptional behavior. Similarly, ECF sigma factors differ structurally from primary sigma factors, for which pausing has been previously reported. Thus, *E. coli* sig70-RNAP pauses at the +6 register, not +7. The authors should perform transcription assays to confirm that initial pausing at the +7 register indeed occurs with the sigE RNAP from Mtb.

We thank the reviewer for this important comment and apologize that the original manuscript did not include *in vitro* transcription data to directly demonstrate initial

transcription pausing in the *Mtb* RNAP- σ^E system. Our structural analysis shows that when the nascent RNA extends from 6 to 7 nt, RNAP undergoes a swivel rotation, which distorts the bridge helix (BH) and stabilizes a paused transcription complex. This structural feature allows us to assign PIC7 as a paused state. We also recognize that *Mtb* RNAP and ECF σ factors differ structurally and functionally from *E. coli* RNAP and primary σ factors, as the reviewer noted.

To directly address this concern, we performed in vitro transcription assays with *Mtb* RNAP- σ^E using a fluorescence-based labeling approach (Wang et al., 2025 *Nature*, 643: 1127-1134). These assays show two distinct RNA species corresponding to 6-nt and 7-nt transcripts (**Extended Data Fig. 1H, I**). The 6-nt band is more prominent, while the 7-nt band is weaker, consistent with previous observations in *E. coli* σ^{70} -RNAP by Duchi et al., *Mol Cell* 2016, 63: 939–950. These data indicate that pausing occurs during the transition from 6-nt to 7-nt RNA, with the 6-nt species representing a post-translocation, pre-paused intermediate, and the 7-nt species representing the paused state (**Page 7, Lines 131-135**).

Together, the cryo-EM structures and the biochemical data validate the assignment of PIC7 as a functional relevant paused state in *Mtb* RNAP- σ^E transcription initiation.

2. CarD has been described as a factor regulating the principal sigma subunit, particularly in ribosomal gene transcription. What is the rationale for including CarD in this study? Is there any published evidence showing that CarD regulates ECF sigma-dependent transcription? Is there any data indicating that CarD is involved in initial pausing? To validate their conclusions, the authors should include in vitro transcription assays demonstrating that CarD activates or modulates sigE-dependent transcription, influences initial pausing, and stabilizes the sigE-RPo complex.

We thank the reviewer for raising this important question. We agree with that CarD has been well described as a factor regulating the principal sigma subunit, particularly in ribosomal gene transcription. We noted that in *Myxococcus xanthus*, CarD can modulate ECF sigma factor activity (Abellón-Ruiz et al., *Environ Microbiol.* 2014, 16:2475-2490). Since *M. xanthus* and *M. tuberculosis* are both actinobacteria and CarD shows high sequence homology between them, this finding suggests that its regulatory role may be conserved. Based on this evidence, we included CarD in our study of *Mtb* σ^E -dependent transcription. In the revised manuscript, we have added this reference to highlight the rationale for including CarD (**References 46**) and also expanded the Introduction to highlight the rationale for investigating CarD in the σ^E

system (**Page 5, Lines 89–93**).

To directly address the reviewer's question regarding whether CarD is involved in initial pausing, we performed *in vitro* transcription experiments (**Extended Data Fig. 1H, I**). These assays show that CarD promotes escape from initial transcriptional pausing: the intensity of the paused 7-nt RNA band is reduced upon addition of CarD. This suggests that CarD assists in pausing escape, likely by interacting with the non-template strand to stabilize the transcription complex and facilitate transcription bubble opening.

Furthermore, FRET-based experiments were conducted to test the stabilization of the RPo complex. As shown in **Fig. 4G**, CarD increases the stability of the σ^E -RNAP-DNA complex, consistent with its proposed role in facilitating productive transcription initiation. Together, these results provide experimental evidence supporting the regulatory role of CarD in ECF σ^E -dependent transcription in *Mtb*.

Specific comments:

1. There is a problem with the experimental setup. First, I found nothing matching the -10 element motif (c/gGTTg/c) for σ^E recognition in the sequence provided by authors. To which *Mtb* promoter corresponds this construct? That information must be provided. Second, they use template with an artificial bubble of 12 nt. Meantime there are only 10 mismatches alternated by two matches. Why? Is there any reason?

We thank the reviewer for raising these important points and sincerely apologize that in our initial scaffold design we did not fully recognize these issues. First, regarding the promoter sequence, we acknowledge that the scaffold we used did not contain the canonical -10 element motif (c/gGTTg/c) for σ^E recognition. Our original design followed the cryo-EM study (Fang et al., *NAR* 2019, 47: 7094-7104), which employed a pre-melted scaffold to capture a chimeric *Mtb* σ^H/σ^E complex. For the artificial transcription bubble, we similarly followed the design (Fang et al., *NAR* 2019, 47: 7094-7104), which contained 12 nt with 10 mismatches interspersed by two matched bases. In our structures, these two bases are nevertheless fully unwound, suggesting that their presence does not influence the bubble architecture or the interactions relevant to pausing (**Page 8, Lines 138-142; Page 9, Lines 169-181**).

We very much agree with the reviewer that addressing these issues is essential for strengthening our conclusions. Therefore, we redesigned scaffolds to incorporate a canonical -10 motif and performed *in vitro* transcription assays, which confirmed pausing at the +7 position (**Extended Data Fig. 1H, I**). We also tested fully

complementary DNA scaffolds to capture the PIC7 state (**Page 9, Lines 169-181**). Although the density was weaker, likely due to scaffold instability or the transient nature of this intermediate, the overall conformation and swiveling observed were consistent with the scaffold-assembled complexes. Together, these additional experiments support that the structures we report represent the paused state (**Extended Data Fig. 4 and Extended Data Fig. 5**).

2. The authors should cite the original publication that introduces and validates the Mango transcription assay used.

We thank the reviewer for the suggestion, and we apologize that the wording in our previous manuscript was not rigorous. The Mango transcription assay used in our study was first introduced and validated by Dolgosheina et al., *ACS Chem Biol* 2014, 9: 2412-2420. We have now added this citation in the Methods section to properly acknowledge the original publication (**Page 30, Lines 597**).

3. The FRET experiment (Figure 3d) is neither informative nor conclusive and could be omitted. Control performed with free DNA is required but missing.

We thank the Reviewer for critical questions and insightful suggestions. We agree that the FRET experiment provides limited mechanistic insight. Therefore, we have moved this experiment to **Extended Data Fig. 7F** rather than the main figure. However, it serves to confirm that our scaffold design is consistent with previous studies. As a control, we have now included measurements with free DNA, showing negligible FRET signal, which supports that the observed signal in **Extended Data Fig. 7F** arises from the assembled transcription complex. We have clarified this in the revised figure legend (**Page 54, Lines 863–864**).

4. page 3 line 47: which “six conserved domains” you mention here ?

We thank the reviewer for pointing this out, and we apologize that the wording in our previous manuscript was not rigorous. The phrase “six conserved domains” was inaccurate. We have corrected the text to refer to “modules” instead, specifying that σ^E contains only σ^2 , σ^4 , and $\sigma^{3.2}$ modules, whereas σ^A contains $\sigma^{1.1}$, $\sigma^{1.2}$, σ^2 , σ^3 , $\sigma^{3.2}$, and σ^4 modules. This more accurately reflects the structural organization of these σ factors. We have corrected the manuscript accordingly (**Page 3, Lines 48–52**).

5. page 4 line 50: usually we call sigma 3.2 a region, not a domain.

We thank the reviewer for the comment, and we apologize that the wording in our previous manuscript was not rigorous. We have revised the manuscript to refer to $\sigma_{3.2}$ as a “region” rather than a “domain,” in line with the standard terminology (**Page 3, Lines 48–52**). We have corrected the manuscript accordingly.

6. page 4 line 59: Could the authors clarify what is meant by “simple domains” ?

We thank the reviewer for pointing this out, and we apologize that the wording in our previous manuscript was not rigorous. We have replaced “simple domains” with “structurally simpler regions” to avoid confusion and better reflect the architecture of σ^E compared to σ^A . We have corrected the manuscript accordingly (**Page 4, Lines 66**).

7. page 4 line 60: provide a reference to a study demonstrating that the affinity of ECF sigma factors to the RNAP core is weaker than that of other sigmas.

We thank the reviewer for this valuable comment and apologize that we could not identify a study directly demonstrating that ECF σ factors have the weakest affinity among all σ families. We have now cited the study by Maeda et al. (Nucleic Acids Res. 2000), which systematically compared the binding affinities of seven *E. coli* σ factors to the RNAP core. This study showed that σ^E along with other alternative σ factors, binds core RNAP weaker than the primary σ^{70}/σ^A . In the revised manuscript, we have therefore phrased this point more precisely as “reduced RNAP-binding stability are the reasons we selected σ^E rather than σ^A for our study.” (**Page 4, Lines 66–68**)

8. page 4 lines 66 – 68: authors wrote that “There is a lack of relevant structural information for Mtb σ^E factor, it is unclear whether σ^E factor folds in the same way as other ECF σ factors and adopts a similar strategy to initiate RNAP transcription.”

That’s an overstatement. In fact, sigE shares high sequence homology with sigH and recognizes the same promoter motifs. The structure of sigH has been solved, and AlphaFold predicts the structure of sigE with high accuracy.

We thank the reviewer for pointing this out, and we apologize that the wording in our previous manuscript was not rigorous. We agree that our original statement was an overstatement. σ^E indeed shares high sequence homology with σ^H and recognizes the same promoter motifs. The structure of σ^H has been experimentally resolved, and

AlphaFold predictions indicate that σ^E adopts a highly similar fold. We have revised the manuscript accordingly to reflect that the structural features of σ^E are expected to closely resemble those of σ^H , while still noting that direct experimental validation remains valuable (**Page 5, Lines 75–77**).

9. page 5 line 80: authors wrote “...we investigated the mechanism of pausing during transcription initiation...”

There is no data on pausing and therefore on the mechanism of pausing.

We thank the reviewer for this important comment. We acknowledge that in the original submission we did not provide direct biochemical evidence of pausing. Structurally, however, our cryo-EM analysis revealed hallmark features of a paused state: at 7-nt RNA, RNAP undergoes a pronounced swivel movement that distorts the bridge helix (BH), a conformational change known to be tightly linked to transcriptional pausing. These features indicate that the PIC7 complexes represent bona fide paused intermediates.

To substantiate this interpretation, we have now performed in vitro transcription assays using *Mtb* σ^E -RNAP on templates containing the canonical -10 element motif. These assays revealed distinct RNA products of 6 and 7 nucleotides (**Extended Data Fig. 1H, I**). The 6-nt transcript is more prominent, consistent with a post-translocation intermediate preceding pausing, while the weaker 7-nt transcript reflects accumulation at the paused state. The weaker intensity of the 7-nt band is expected, since the pause is transient and the majority of complexes either remain at 6 nt or proceed into elongation.

Taken together, the combination of structural signatures (swiveling and BH distortion) and functional data (discrete 6-nt and 7-nt RNAs) confirm that our structures capture paused initiation complexes. We have revised the Results section (**Page 7, Lines 131-135**) to include these new experimental findings.

10. page 5 lines 85-86: authors wrote “We revealed that RNA polymerase pauses between transcript lengths of 6 and 7 nucleotides...”. As I explained above, there is no any data showing that RNAP indeed pauses at +7 register.

We appreciate the reviewer’s concern, and we apologize that our previous description was not sufficiently supported by biochemical evidence. In the revised manuscript, we have now included in vitro transcription assays with *Mtb* σ^E -RNAP on

a template containing the -10 element motif (**Extended Data Fig. 1H, I**). These assays show RNA products corresponding to 6- and 7-nucleotide lengths, with the 6-nt band more prominent and the 7-nt band weaker, consistent with previous observations in *E. coli* σ^{70} -RNAP by Duchi et al., *Mol Cell* 2016, 63: 939–950. These data indicate that pausing occurs during the transition from 6-nt to 7-nt RNA, with the 6-nt species representing a post-translocation, pre-paused intermediate, and the 7-nt species representing the paused state. Our structural data also reveal that RNAP undergoes a conformational change associated with pausing when the RNA extends from 6 to 7 nucleotides. Together, the structural and biochemical data provide support for our conclusion that RNAP pauses at the 6 to 7 nt register in this system.

11. page 5 lines 88 – 91: Authors wrote “We also found that CarD binds to non-template ssDNA in the IC complex to stabilize the transcription bubble, whereas it interacts with template ssDNA in the PIC complex to stabilize the paused transcriptional conformation.”

There is no experimental evidence demonstrating that CarD stabilizes the transcription bubble. If authors wish to prove it, they should perform KMnO₄ probing on RPo complexes with and without CarD. Also, there is no data showing that CarD stabilizes “paused transcription conformation”. To prove it, authors should perform in vitro transcription kinetics assay with and without CarD.

We thank the reviewer for pointing out that there is no direct experimental evidence demonstrating that CarD stabilizes the transcription bubble. We apologize that our previous description was not sufficiently rigorous. To address this, we performed FRET experiments using fluorescently labeled transcription bubble scaffolds (following Cordes et al., *Biochemistry* 2010, 49:9171-9180), which show that CarD promotes bubble stability in the σ^E -RNAP system (**Fig. 4G**). Although KMnO₄ probing was suggested, we were unable to perform this assay due to restrictions on piperidine in China. The FRET approach serves as a widely used alternative to monitor transcription bubble dynamics.

We also thank the reviewer for raising the concern regarding the statement that CarD stabilizes the “paused transcription conformation.” We apologize that this description was imprecise. Based on our structural and biochemical data, a more accurate interpretation is that in the initiation complex (IC), CarD interacts with the template strand to maintain the open complex. In the paused initiation complex (PIC), CarD binds the non-template strand, which stabilizes the transcription bubble and facilitates pause escape. This conclusion is supported by in vitro transcription assays

showing that CarD decreases the intensity of the +7 nt paused RNA band (**Extended Data Fig. 1H, I**). We have revised the Results section (**Page 17, Lines 341–347**) to incorporate these findings and clarify the role of CarD.

12. page 6 lines 101-104: authors refer to the studies on *E. coli* sig70-RNAP reporting pausing at +7 register. This is not relevant because they use a different RNAP and a structurally distinct sigma factor.

We thank the reviewer for this point. We have performed new in vitro transcription experiments using *Mtb* σ^E -RNAP, which confirm that initial transcription indeed pauses at the 6 to 7 register (**Extended Data Fig. 1H, I**). Although *E. coli* σ^{70} -RNAP also pauses at this position, our results demonstrate that this pause is conserved in *Mtb* despite structural differences between the RNAPs and sigma factors. This supports the relevance of our structural analyses of the paused complexes.

13. page 9 lines 159 – 160: I see no evidence showing that CarD stabilized the non-template ssDNA of the transcription bubble.

We thank the reviewer for this thoughtful comment. In the paused transcription complex, we observed that CarD interacts with the non-template strand to stabilize the transcription bubble. Using a fluorescence-labeled transcription bubble assay (Cordes et al., *Biochemistry* 2010, 49: 9171-9180), we observed that CarD promotes transcription bubble formation with σ^E , indicating stabilization of the RPo (**Fig. 4G**). Consistently, in vitro transcription assays show that CarD decreases the intensity of the +7 nt paused RNA band, supporting its role in stabilizing the bubble and facilitating pause escape (**Extended Data Fig. 1H, I**). Together with our structural observations of CarD binding to the non-template strand, these data support its role in transcription bubble stabilization and pause modulation.

14. page 9 lines 163-165: authors wrote “It has been known for decades that RNAP adopts a pausing state when the nascent RNA extends to 6 to 7 nucleotides.” If authors refer to the initial transcription pausing by *E. coli* sig70-RNAP, it has been published in 2016. It’s not “for decades”.

We thank the reviewer for pointing this out, and we apologize that the wording in our previous manuscript was not rigorous. We agree that describing this as “for decades” was an overstatement. The initial transcription pausing by *E. coli* σ^{70} -RNAP

was first reported in 2016 (Duchi et al., *Mol Cell* 2016, 63: 939-950). We have revised the text accordingly to state that this phenomenon was first described in 2016 (**Page 11, Lines 218-220**).

15. page 11 lines 196-197: authors wrote "... the $\sigma^{3.2}$ domain of σ^E stimulates pausing through its insert into RNA exit channel, forming a roadblock that promotes RNAP swiveling and interrupts nucleotide addition."

As explained above, there is no biochemical data supporting this conclusion.

We thank the reviewer for this important comment and apologize that our initial manuscript did not include biochemical evidence to support this conclusion. We observed that the $\sigma^{3.2}$ region of σ^E inserts into the RNA exit channel and forms a steric roadblock. As the RNA extends, the 5' end of the nascent transcript clashes with the $\sigma^{3.2}$ region, and this steric interference promotes swiveling of RNAP, ultimately leading to transcriptional pausing. To validate this mechanism, we have now performed in vitro transcription assays (**Extended Data Fig. 1H, I**), which confirm that *Mtb* σ^E -RNAP pauses at the 7-nt stage. These biochemical data, together with our structural observations, provide support for the revised description of $\sigma^{3.2}$ -mediated pausing. We have updated the manuscript accordingly (**Page 13, Lines 251-255**).

16. page 13 lines 241 – 242: authors wrote "The FRET assay validated this conformational change, consistent with previously reported changes in *E. coli*." The ensemble FRET measurements performed by authors do not include essential controls which allows to validate it.

We appreciate the reviewer's comment, and we apologize that the wording in our previous manuscript was not rigorous. In the revised manuscript, we have clarified that a DNA-only control was included in our ensemble FRET measurements, allowing us to exclude the possibility that the observed signal originated from free DNA rather than RNAP-bound complexes (**Extended Data Fig. 7F**). While we acknowledge that ensemble FRET has inherent limitations compared with single-molecule approaches, our results are consistent with previously reported conformational changes in *E. coli* RNAP. We have revised the relevant text to clarify the controls and the scope of interpretation (**Page 15, Lines 293-298**).

17. page 13 lines 253 – 255: authors wrote "This minimal interface suggests a transient

association mechanism at the transcriptional checkpoint, distinct from stable binding.”

Its not clear what do you mean under “transcriptional checkpoint” here ?

We thank the reviewer for raising this point. In our manuscript, by “**transcriptional checkpoint**” we specifically refer to the decision point during initiation when RNAP can either pause or proceed into productive elongation. At this stage, σ^E engages with RNAP through a minimal and transient interface, which differs from the more stable associations observed in elongation complexes. To avoid ambiguity, we have revised the text to clarify this meaning (**Page 16, Lines 309-312**). The revised sentence now reads: *“This minimal interface suggests a transient association mechanism at the transcriptional checkpoint, i.e., the stage where RNAP either pauses or escapes into productive elongation, distinct from stable binding.”*

18. Extended data Figure 1j. The EMSA performed with the bubble template is not informative and adds nothing. It shows that there is binding, but there is nothing to prove that there is a complex with sigE-CarD-RNAP. RNAP core alone would bind and shift this template perfectly well. Also, I do not understand why so huge amounts of RNAP (18 mg) were used ? Also, final concentration of proteins is not mentioned in the Methods section, while it should be.

We thank the Reviewer for critical questions and insightful suggestions. In the revised manuscript, we have repeated the EMSA experiments with appropriate controls (σ^E , CarD, RNAP core, and σ^E -RNAP-CarD) to better illustrate the contributions of each component. These new data are now included in **Extended Data Figure 1G**. we apologize for the mistake in reporting the amount of RNAP, which was incorrectly listed as 18 mg; the correct amount is 18 μ g. In addition, we have updated the Methods section to include the final concentrations of proteins used in the EMSA assays for clarity and reproducibility.

19. Extended data Figure 1h . Is it expected that CarD will form a stable complex with sigE (peak2) ? How can it contribute to regulation ?

We appreciate the reviewer’s careful observation. Peak 2 does not represent a stable CarD- σ^E binary complex. Rather, it arises from the co-elution of excess CarD and σ^E added during assembly, which were not completely resolved during size-exclusion chromatography. We have clarified this point in the revised legend of **Extended Data Fig. 1E** to avoid confusion (**Page 43, Lines 775-776**).

Although CarD does not form a stable complex with σ^E alone, it contributes to regulation by stabilizing the σ^E -RNAP-DNA complex. As shown by our FRET experiments, CarD promotes transcription bubble stability (**Fig. 4G**), and our in vitro transcription assays further demonstrate that CarD reduces the accumulation of the +7 paused RNA, facilitating pause escape (**Extended Data Fig. 1H, I**). These results suggest that CarD functions as a transient co-factor: rather than binding stably to σ^E in isolation, it interacts with the transcription complex in a dynamic manner to modulate pausing and initiation efficiency.

20. Figure 3. There is a mismatch between Fig.3 panel lettering (panels g, h) and Fig. 3 legend (panels j,k).

We thank the reviewer for pointing this out and apologize for the oversight in the panel labeling. The panel lettering and corresponding figure legend in **Figure 3** have now been corrected so that they match accurately.

21. EMSA in Figure 3g is not informative. Which DNA was used ? Was there a competitor added ? What is a rational in doing EMSA with sigE alone ?

We thank the reviewer for this helpful comment. In the original EMSA, we used a pre-melted DNA scaffold (**Extended Data Fig. 1D**) that was identical to the one employed for assembling cryo-EM complexes, in order to maintain consistency between the biochemical and structural experiments. No competitor DNA was included at that stage, since our purpose was to test the direct binding capacity of σ^E and its mutant under simplified conditions. We also examined σ^E alone because when RNA polymerase is present, σ^E is largely embedded within the polymerase structure, and simple point mutations may not produce detectable differences in DNA binding compared to wild-type. Measuring σ^E on its own allowed us to assess the intrinsic DNA-binding properties of the wild-type and mutant proteins

To address the reviewer's concerns, we have now redesigned the EMSA experiments. In the revised assays, we used fully complementary DNA (**Extended Data Fig. 1H**) containing the canonical -10 motif (c/gGTTg/c) and compared wild-type and mutant σ^E both alone and in complex with RNAP. This new setup enables us to assess intrinsic DNA-binding while also evaluating the effect of RNAP on complex formation. The revised experiments, now included in **Fig. 3G and Extended Data Fig. 8E**, provide clearer and more informative results that complement the structural analyses.

22. Figure 4g. A control without CarD is missing. I do not see how this experiment reveals CarD cooperativity .

We thank the Reviewer for critical questions and insightful suggestions. In the revised experiments, we included a control without CarD to establish the baseline transcription behavior. Compared with this control, addition of wild-type CarD significantly enhances transcription activity, and the CarD mutant also increases activity but to a lesser extent, indicating partial retention of function (**Fig. 4H**). These results demonstrate the cooperative effect of CarD in σ^E -dependent transcription.

Furthermore, in vitro transcription pausing assays (**Extended Data Fig. 1H, I**) and FRET-based bubble stability measurements (**Fig. 4G**) both show that CarD cooperates with σ^E to stabilize the transcription complex and facilitate escape from the paused state. Together, these data provide multiple lines of evidence that CarD acts cooperatively with σ^E to modulate transcription, supporting our conclusions.

23. Overall, manuscript should be rewritten in more concise form. Some sections from Results may go to Introduction (e.g. p12 lines 222- 227, p14, lines 261-265). Discussion is too excessive. Speculations on NusG and transcription -translation coupling are irrelevant to the current study.

We thank the Reviewer for critical questions and insightful suggestions. We have moved the sections highlighted (p12, lines 222-227; p14, lines 261-265) into the Introduction (**Page 4, Lines 59-65; Page 5, Lines 85-89**). We have also consolidated similar content in the Discussion and streamlined the text to make the manuscript more concise. The sections on NusG and transcription-translation coupling have been removed, as they are not directly relevant to the present study. These revisions make the manuscript more concise while retaining important mechanistic discussions relevant to transcription initiation, CarD function, and σ^E retention.

We sincerely thank the reviewers and editor for their thoughtful and constructive comments, which have helped us substantially improve the manuscript. In this revised version, we have carefully addressed all remaining issues through additional experiments, clearer descriptions, and refined interpretations.

In this revision, we have:

- Performed new in vitro transcription assays with promoter scaffolds identical to those used for cryo-EM, which confirmed a transcriptional pause at the +6/+7 RNA register.
- Expanded the FRET analyses by providing detailed efficiency calculations and correlating the results with predicted structural transitions.
- Conducted a fluorescence-based assay demonstrating CarD-mediated stabilization of the σ^E -RPO complex.
- Clarified figure legends, improved terminology and labeling, and removed speculative statements to enhance clarity and precision.
- Revised the Results and Discussion sections to emphasize the transient and regulatory nature of pausing and the modulatory role of CarD.

Referee's remarks are shown in black and our responses are in blue. All modifications in the revised manuscript are highlighted in red.

REVIEWER COMMENTS

Reviewer #1 (Remarks to the Author):

The authors have satisfactorily addressed all my previous concerns in the revised manuscript. I have no further comments and fully support the publication of this study.

We thank the Reviewer for the generous and encouraging comments. We are delighted that our revisions have addressed the reviewer's concerns and that our work is now fully supported for publication.

Reviewer #2 (Remarks to the Author):

The authors have made substantial efforts to clarify many points in their response and the revised manuscript, and addressed most issues raised in the reviews. The manuscript is clearly interesting and making an important contribution. However, some key questions and revisions remain:

Re my main comment #1: the authors need to caveat their observation with the fact that scrunching in their complexes is clearly very different from complexes prepared through an actual reaction. The fact that other works had used this strategy does not take away the fact that they had the same caveat; I understand the necessity to get stable states, but the states may be different from the ones formed during the actual reaction, and this needs to be explicitly stated in the ms.

We thank the reviewer for this insightful comment emphasizing the distinction between our scrunched complexes and those formed during actual transcription reactions. We agree that our structures represent stabilized intermediates, which may differ in subtle ways from the transient states formed under native reaction conditions.

We have now explicitly stated this limitation in the Results section (**page 15, lines 291–300**) and further discussed it in the Discussion section (**page 20, lines 388–398**), emphasizing that although such stabilized complexes have been widely used in previous structural studies, they may not fully recapitulate the dynamic nature of scrunching during active transcription.

Re my main comment #5. The FRET assay remains poorly described. What is the FRET efficiency of the various samples? What is the prediction from the structure? Why is the FRET efficiency different only for RNA7? And not for RNA6 or RNA8? This assay needs a proper introduction, predictions, description, and discussion, as well as a better illustration.

We thank the reviewer for this insightful comment and apologize for the insufficient description in the previous submission. In the current revision, we have defined FRET efficiency as the normalized change in Cy3 fluorescence relative to the DNA-only control. The detailed calculation procedure is now described in the Methods section (**page 30, lines 602–607**) and added corresponding explanations in the legend of **Extended Data Fig 7F (page 56, lines 910–919)**. In the Results section (**page 16, lines 305–315**), we further describe how the measured FRET efficiencies correspond to the structural transitions observed during RNA extension. Specifically, structural analysis suggests that elongation of the RNA induces a steric clash with the $\sigma^{3.2}$

region, resulting in conformational rearrangements that elevate the upstream DNA and thus reduce FRET efficiency. At the 7-nt RNA stage, the complex exists as a mixture of paused and non-paused states, resulting in a relatively lower FRET efficiency. In contrast, RNA5 and RNA6 have not yet reached the $\sigma_{3.2}$ region and therefore show relatively higher FRET efficiencies, while RNA8 and RNA9 likely overcome the $\sigma_{3.2}$ barrier or undergo backtracking, resulting in increased FRET again. These interpretations are consistent with our structural observations, and we have clarified them in the Results section (**page 16, lines 305–315**). In addition, the corresponding FRET efficiency data have been added to **Extended Data Fig. 7F** to illustrate these trends.

We thank the reviewer for this insightful comment and apologize for the insufficient description and lack of clarity in the previous submission. To address the reviewer's suggestion for a clearer introduction, predictions, description, and discussion of the FRET assay, we have comprehensively revised the relevant text throughout the manuscript. Specifically, we have introduced the assay rationale and FRET efficiency calculation in the Methods section (**page 30, lines 602–607**). We have also included structure-based predictions of FRET efficiency changes and described the correspondence between structural rearrangements and fluorescence signals in the Results section (**page 16, lines 305–315**). Furthermore, we have discussed the limitations of the current bulk FRET measurements and the need for further validation in the Discussion (**page 29, lines 498–504**). In addition, we have added a schematic illustration and explanatory notes in **Extended Data Fig. 7** to clarify the FRET mechanism, summarize the experimental design, and visualize the relationship between structural rearrangements and FRET efficiency changes.

Re my minor comment #7: Fig 1I needs to be annotated so we can understand what the "SigE" lane is. Further, if the sigE lane is for the run-off reaction, then the band that accumulates is the 6-mer, not the 7-mer (as in Duchi et al Mol Cell 2016 for the E.coli system), hence the pause (i.e., the point in synthesis where the conversion from n to n+1 occurs) occurs at the 6-mer level. As a result, the discussion and model need to adapt to accommodate the functional data.

We thank the reviewer for the helpful comment and apologize for the unclear labeling in the previous submission. In the current revision, we have now corrected the annotation in **Extended Data Fig. 1I** and clarified in the figure legend that the "SigE" lane represents the run-off transcription reaction.

We thank the reviewer for this careful observation and apologize that our previous

revision did not sufficiently integrate the structural and biochemical views of pausing at the 6-nt to 7-nt stage. Transcriptional pausing during initial transcription is inherently dynamic, and complexes are expected to sample multiple, closely related intermediates rather than a single discrete state. In our cryo-EM analysis, the paused conformation that we capture at a nominal RNA length of 7-nt adopts a half-translocated architecture: the RNA has reached 7-nt, whereas the template DNA in the active site remains in a pre-translocated register equivalent to the 6-nt position (**Fig. 1F**). This half-translocated arrangement is a well-recognized hallmark of paused RNAP complexes. Therefore, we propose that pausing occurs during the transition between the 6-nt and 7-nt RNA lengths, rather than being confined to a specific RNA length of either 6-nt or 7-nt.

Consistent with this, our *in vitro* transcription assays show two species at 6-nt and 7-nt, with the 6-nt band more prominent and the 7-nt band weaker. Taken together, the structural and biochemical data suggest that pausing occurs during the transition from 6-nt to 7-nt RNA synthesis, rather than being strictly confined to a single RNA length. In the revised manuscript, we therefore do not assign the pause to “6-nt” or “7-nt” alone, but instead describe it as a checkpoint spanning the 6-nt to 7-nt transition. This interpretation is in line with the kinetic analysis by Duchi *et al.* (*Mol Cell* 2016, 63: 939–950), where extension from 6-nt to 7-nt was found to be rate-limiting and elongation beyond 7-nt proceeded more rapidly. We have revised the relevant text in the Results (**page 13, lines 251–253**) and Discussion (**page 24, lines 476–479**) to reflect this view of a dynamic 6-nt to 7-nt checkpoint rather than a pause assigned to a single register.

I also think that the term “allosteric” is still a bit confusing, and best to avoid. The term “RNA-induced” seems both adequate and appropriate.

We sincerely thank the reviewer for the helpful suggestion and apologize for the inaccurate terminology used in our previous version. The term “allosteric” did not precisely capture the mechanistic nature of the process we describe. We appreciate the reviewer’s clarification and fully agree that “RNA-induced” provides a more accurate and appropriate description of the conformational changes observed. Accordingly, we have revised the manuscript to replace “allosteric” with “RNA-induced” throughout the relevant sections to ensure terminological accuracy and conceptual clarity.

Reviewer #3 (Remarks to the Author):

While the revised manuscript has been improved my principal concerns have not been resolved.

1. My point was that the authors provided no evidence for pausing at the +6/+7 registers on the promoter templates used for cryo-EM. In the revised version, the authors have added a run-off transcription assay (Extended Data Fig. 1H,I). However, upon reading, it appears that this assay was performed using a different template and under different conditions: including a different -10 sequence, dsDNA bubble instead of a mismatch bubble, and de novo initiation instead of RNA primers. Since pausing is highly dependent on the DNA template architecture, I do not see how this experiment can support the structures presented by the authors. In essence, there are structures and there is a functional assay, but they are disconnected.

Finally, a single run-off data point does not demonstrate pausing at +6/+7. Although there are weak bands corresponding to 6-nt and 7-nt RNAs, these could simply represent abortive products. The only proper way to demonstrate pausing is to monitor the kinetics of promoter escape: if the +6/+7 bands first increase and then disappear, one could then conclude that a pause occurs.”

Furthermore, in the Figure legend, authors should better explain what is shown on the gel. What is in the “control” line ? How lines with 6nt-pCp AF647 and 7nt-pCp AF647 were obtained ? Are these just RNA primers alone or in complex with RNAP ?

We thank the reviewer for this insightful comment and for emphasizing the importance of directly correlating biochemical assays with the structural analysis. In the current revision, we have performed additional in vitro transcription assays using a pre-melted promoter DNA scaffold identical to that used for cryo-EM sample preparation (**Extended Data Fig. 1J, K**). Because fixed RNA primers could bias the detection of potential paused intermediates by predefining the transcript length, we instead employed de novo transcription reactions to more accurately capture all RNA species generated during initiation. To maintain full consistency with the cryo-EM samples, the DNA scaffolds used in these assays were relatively short, which resulted in run-off transcripts migrating close to the paused RNA species on the gel (**Extended Data Fig. 1 K**). These new time-course experiments were designed to provide kinetic evidence of pausing at the +6/+7 registers, and the results indeed show a gradual reduction in the intensity of the 6-nt and 7-nt RNA bands as the reaction proceeds, particularly between 30 and 60 minutes (**Extended Data Fig. 1K**). Because transcription initiation pausing is inherently coupled with abortive release, the reaction involves overlapping processes that make it difficult to obtain a sharply defined kinetic

trend. We also tested shorter reaction intervals, but the RNA yield was too low and the bands became less discernible. Nevertheless, the overall pattern supports the interpretation that a transient paused intermediate exists during early transcription initiation. Moreover, σ^E +CarD reactions show reduced accumulation of paused products, consistent with the results obtained using the fully matched promoter template (**Extended Data Fig. 1I, K**).

In the current revision, we have revised the figure legends (**page 45, lines 813–833**) to clarify the composition of the “control” lanes and the preparation of the AF647-labeled 6-nt and 7-nt RNA markers. The “control” lanes represent reactions performed without promoter DNA, while all other components were identical to the σ^E +CarD reaction, allowing us to assess background RNA signals. The 6-nt-pCp–AF647 and 7-nt-pCp–AF647 markers were prepared by labeling synthetic RNA oligonucleotides of 6 and 7 nucleotides in length with pCp-AF647, following the procedure described in **Extended Data Fig. 1H**. All RNA species analyzed in these assays represent RNA oligonucleotides alone rather than RNAP-RNA complexes, as fluorescence labeling was performed after TRIzol extraction of RNA products from the completed reactions prior to gel analysis (**Extended Data Fig. 1H, J**). Therefore, no RNA primers were present in complex with RNAP, and the labeled RNA alone was analyzed on denaturing urea PAGE gels. Additionally, we have slightly adjusted the phrasing in the Results section (**page 18, lines 360–363**) to state that the observed structural features are “consistent with the trends observed in our in vitro transcription assays,” reflecting the transient and partially overlapping nature of pausing and abortive events.

2. In their response letter authors explain the rationale for using CarD by analogy with CarD from *Myxococcus xanthus* which has been shown to regulate ECF sigma transcription. While CarD is indeed a conserved protein, *Myxococcus xanthus* is not an Actinobacteria, as claimed by authors, and such comparison is inappropriate. See: <https://www.ncbi.nlm.nih.gov/Taxonomy/Browser/wwwtax.cgi?id=34>

We thank the reviewer for this valuable correction and apologize for the inaccurate description in the previous submission. In the current revision, we have removed the taxonomic reference and rephrased the statements to avoid any direct comparison between *Myxococcus xanthus* and *Mycobacterium tuberculosis* (**page 5, lines 86–89**). The revised text now simply notes that CarD has been reported to modulate transcription mediated by an ECF σ factor in *M. xanthus* (**Ref. 46**), and highlights that its potential role in σ^E -dependent transcription in *M. tuberculosis* has not been characterized before. This change accurately reflects the published evidence and avoids overinterpretation.

3. Its not clear how CarD can stimulate pause escape if it stabilizes interaction with the ssDNA of transcription bubble. In that case one would expect an opposite: CarD will stabilize pause.

We thank the reviewer for this important comment. Indeed, previous studies have demonstrated that CarD stabilizes the open promoter complex (R_{Po}) and prevents bubble collapse. These findings are consistent with our observation that CarD enhances σ^E -dependent transcription initiation.

Our results indicate that during the transition from 6-nt to 7-nt RNA synthesis, CarD reduces the lifetime of the paused state (**Extended Data Fig. 1I, K**). In our structural analysis, we observe that CarD binds to the template strand in the initiation complex, while it binds to the non-template strand during the paused state (**Fig. 4A-D**). In both scenarios, CarD serves to stabilize the transcription bubble, although the binding mode differs. Specifically, during initiation, CarD stabilizes the transcription bubble to facilitate the unwinding of the DNA duplex and prevent collapse. During the paused state, CarD stabilizes the scrunched transcription bubble, helping to ensure the transition from the paused state to productive elongation. This dual role is consistent with our biochemical experiments, which show that CarD reduces the intensity of the 6-nt and 7-nt paused bands (**Extended Data Fig. 1I, K**). Therefore, we propose that CarD's function in transcription initiation is to stabilize the transcription bubble throughout, first enabling efficient DNA opening and later stabilizing the scrunched bubble to support the transition to elongation, rather than prolonging the pause.

We have updated the Results (**page 19, lines 360–372**) and Discussion (**page 24, lines 480–485**) sections to clarify this dual functionality of CarD, emphasizing its role in both initiation and pausing, and to better explain how it stabilizes the transcription bubble in a conformation conducive to transcription elongation.

4. Fig. 4G FRET assay does not directly demonstrate any effect on bubble 'stability'. It only shows an increase in fluorescence, which the authors interpret as an effect on stability. If the authors wish to make a stronger claim regarding R_{Po} stability, they should perform a dissociation kinetics study, as I suggested during the first round of revision. This analysis could also be carried out using FRET. As a more simple fluorescent assay they can use promoter with Cy3 probe in transcription bubble (e.g. DOI: 10.1093/nar/gkw577)

We thank the reviewer for this constructive suggestion and for pointing us to the relevant reference (DOI: 10.1093/nar/gkw577). Following this recommendation, we

have re-examined the effect of CarD on RPo formation using a fluorescence-based kinetic assay inspired by this method. Although a stopped-flow instrument was not available to achieve millisecond resolution, we performed a time-resolved fluorescence assay with second-level temporal resolution (**Fig. 4G**). The results consistently show that the presence of CarD leads to a higher plateau compared to the RNAP- σ^E -only complex, indicating that CarD stabilizes the σ^E -RPo complex and enhances its steady-state accumulation. This new dataset has been included in the revised **Fig. 4G** and described in the Methods (**page 31, lines 617–625**) and Results (**page 19, lines 365–372**) sections.

5. p4 lanes 65-71: speculations about weak affinity of sigE to Mtb RNAP are unwarranted. You cannot, and should not, draw conclusions about the relative affinities of sigA versus ECF sigma factors for *M. tuberculosis* (Mtb) RNAP based on the corresponding affinities of sig70 versus ECF sigma factors in *E. coli*. Numerous studies over the past decades have clearly shown that Mtb RNAP behaves differently from *E. coli* RNAP; for example, sigA exhibits only weak affinity for Mtb RNAP in the absence of RbpA.

We thank the reviewer for raising this point and fully agree that inferring σ^E -RNAP binding affinity in *Mtb* based on published data from *E. coli* was inappropriate. We apologize for this speculative interpretation. Since this inference is not essential to our conclusions, we have removed the statement regarding relative binding affinity between σ factors (**page 4, lines 63–67**). The revised text now focuses only on the rationale for choosing σ^E — its simpler domain architecture and suitability for capturing initiation intermediates — without making assumptions about RNAP- σ binding strengths.

6. p. 17 “line 343 “...CarD ... facilitates pausing at the 6-7 nt RNA stage”.

It contradicts to your pervious conclusion that CarD suppresses pausing.

We apologize for the confusion caused by the previous wording. We thank the reviewer for carefully noting this inconsistency. In our revised manuscript, we have clarified that CarD does not directly facilitate pausing but rather modulates the paused state. Our structural analysis indicates that CarD binds to the template strand during initiation, and to the non-template strand during pausing, playing a stabilizing role in both cases. Specifically, in the initiation state, CarD prevents premature collapse of the transcription bubble, whereas in the paused state, it stabilizes the scrunched

transcription bubble, facilitating the transition from pausing to elongation. This dual role is consistent with our biochemical data, which show that in the presence of CarD, the 6-nt and 7-nt paused bands become weaker (**Extended Data Fig. 1H, I**), supporting the idea that CarD reduces the duration of pausing and promotes escape from the paused state.

In the revised manuscript, we have made the necessary changes to the Results (**page 19, lines 365–372**) and Discussion (**page 24, lines 480–485**) sections to ensure our interpretation aligns with the structural and biochemical data. This correction now reflects that CarD's role is not to directly facilitate pausing, but rather to modulate the paused state and promote transition into productive elongation.

We sincerely thank the reviewers and the editor for their thoughtful and constructive comments, which have helped us further improve the clarity and rigor of the manuscript. In this revised version, we have carefully addressed all remaining concerns through additional clarification, improved connections between the structural and biochemical data, and a more conservative interpretation of the initiation pausing mechanism.

In this revision, we have:

- Removed the ensemble FRET dataset, as technical limitations made its interpretation uncertain and the data were not required for our mechanistic conclusions.
- Clarified the correspondence between the structural and biochemical complexes, including how the matched +1 to +7 scaffold relates the structures to de novo initiation, and added a concise explanation of the inherent limitations of de novo assays at the 6–7 nt stage.
- Refined the presentation of the initiation-paused intermediate and the role of CarD throughout the Abstract, Introduction, Results, and Discussion, ensuring that our interpretations rely strictly on structural and biochemical evidence and avoiding overstating the contribution of any single factor.

Referee's remarks are shown in black and our responses are in blue. All modifications in the revised manuscript are highlighted in red.

Reviewer #2 (Remarks to the Author):

The authors very diligently addressed most of my remaining comments, and the manuscript is now very strong.

My remaining question rests with the FRET assay. The included description and schematics are very helpful. The authors acknowledge that the limitation of their current FRET (at the end of the Discussion) but I agree with them that there is an ensemble FRET signature that is significant and most likely interpretable. However, I also see that they suggest that the FRET increase in the RNA8 and RNA9 samples is because “RNA may overcome the σ 3.2 barrier or undergo limited backtracking” . While the backtracking hypothesis will indeed provide such signals, it will be important that the authors point to where in the paper they show that overcoming the σ 3.2 barrier restores proximity by the upstream DNA swiveling back (as suggested in the text and in the schematic of RNA8/9 in Fig. 7F.); it was not clear to me.

The authors should also discuss how a recent smFRET paper on the length dependence of σ 3.2 displacement (DOI: 10.1093/nar/gkaf857) aligns with their results and interpretation.

We sincerely thank the reviewer for the positive assessment and for the helpful comments regarding the FRET experiments. After carefully evaluating the ensemble FRET dataset, we concluded that its technical limitations, including sensitivity of Cy3 intensity to local environment and the difficulty of interpreting signals at longer RNA lengths, make it less reliable than our structural and biochemical evidence. To avoid potential overinterpretation and because the data are not essential for our mechanistic conclusions, we have removed the FRET panel from the revised manuscript.

We also attempted to determine structures for complexes containing 8 or 9 nt RNAs but were not able to obtain reconstructions of sufficient quality. Based on the paused complex that we resolved and on established elongation-state structures, we infer that progression toward elongation restores the upstream DNA and active site to a non-swiveled, catalytically competent configuration, which is consistent with the interpretation suggested by the reviewer.

We are grateful to the reviewer for pointing out the recent smFRET study (Nucleic Acids Res, 2025, 53: gkaf857). This work supports the initiation pause centered on the 6 to 7 nt transition and aligns well with our structural observations. We now cite this study in the revised manuscript (**Reference 64**) and briefly discuss how it supports our interpretation of the initiation pause (**Page 22, Lines 429-431**).

Reviewer #3 (Remarks to the Author):

I appreciate the efforts made by the authors to improve their manuscript. However, major concerns listed below remain unresolved.

1. Lack of Evidence for Pausing at +6 or +7 and no effect of CarD on pausing

We thank the reviewer for the thoughtful assessment. We address the overarching concern below by responding point-by-point to each of the specific issues raised.

The new experimental data presented in the revised version do not support the central conclusions regarding transcriptional pausing at the +6 or +7 registers. In fact, the new supplementary Fig. 1K clearly shows no pausing. The faint band that the authors interpret as a pause at +6/+7 can just as well represent an abortive RNA product.

We thank the reviewer for the thoughtful comments. Multiple studies have shown that early transcription proceeds through a scrunched, transiently paused intermediate that can give rise to both productive extension and abortive release, with both outcomes originating from the same underlying state rather than from two mechanistically distinct intermediates (Nucleic Acids Res, 2025, 53: gkaf857; Mol Cell, 2020, 79:797-811; Mol Cell 2016, 63: 939–950; Nat Commun 2018, 9: 1478). Within this framework, the appearance of 6–7 nt RNA products in our de novo assays, regardless of whether some portion is abortive or productive, reflects transient occupancy of this early paused intermediate. In the revised manuscript, we now explicitly acknowledge this limitation in the Discussion (**Page 22, Lines 445-453**) and have refined the wording in the Abstract (**Page 2, Lines 18-21, Lines 25-27**), Introduction (**Page 6, Lines 100-102; Page 7, Lines 117-118**), Results (**Page 10, Lines 177-179**), and Discussion (**Page 21, Lines 423-431; Page 25, Lines 489-491, Lines 503-505**) so that our interpretation does not rely on separating paused versus abortive 6–7 nt species. These adjustments improve clarity and strengthen the rigor of the manuscript.

While it is possible that one of the reported structures corresponds to a PIC, the structures remain disconnected from the biochemical assays. In the structures, the 7-nt RNA primer anneals to positions –7 to –1 of the template DNA. In the transcription assay, under de novo initiation, a 7-nt nascent RNA would instead anneal to positions +1 to +7. Thus, the sequence contexts are different, and the complexes formed in de

novo transcription initiation are not equivalent to those used for structure determination.

We thank the reviewer for recognizing that our structures represent an initiation-paused state, a key regulatory intermediate during transcription initiation. Our study, to our knowledge, is the first to resolve this intermediate at high resolution. We appreciate the reviewer's concern regarding the relationship between the structural and biochemical complexes. To directly address this, we determined structures using a fully matched scaffold in which the 7 nt RNA anneals to positions +1 to +7, identical to the register generated during de novo initiation (**Extended Data Fig. 1H and 1I**). As shown in **Extended Data Fig. 4 and Fig. 5**, this matched complex exhibits the same paused architecture as the pre-melted -7 to -1 scaffold. The use of a pre-melted RNA was solely to obtain complete nucleic-acid density, a widely applied approach in bacterial RNAP structural studies (Nucleic Acids Res 2019, 47: 7094-7104; Nat Commun 2019, 10:1153). Although de novo transcription on the pre-melted scaffold does not reproduce the -7 to -1 register used for structural determination, it nevertheless generates prominent 6-7 nt RNA species (**Extended Data Fig. 1J and 1K**). Together with the observation that the fully matched +1 to +7 scaffold adopts the same paused architecture as the pre-melted scaffold, this supports the use of the pre-melted scaffold as a structurally valid approach for capturing the initiation-paused intermediate. In the revised manuscript, we have further clarified in both the Results (**Page 7, Lines 127-137**) and the Discussion (**Page 22, Lines 432-444**) how the matched and pre-melted scaffolds validate each other and how they relate the structural intermediates to the de novo assays.

It is unclear why the authors did not perform one straightforward experiment that would support their claims. They have at hands two types of complexes assembled on synthetic templates: one with a 6-nt RNA (productive) and one with a 7-nt RNA (paused + productive). They only need to add NTPs to these complexes and examine how RNAP extends the 6-nt and 7-nt RNAs. If, as they propose, IC7 is paused whereas IC6 is not, then a clear difference in kinetics should be observed, with a significant fraction of the 7-nt RNA failing to extend. Because the kinetics are rapid, the experiment should be conducted within a time range of less than 600 seconds (see PMID: 35318334; PMID: 33380428, PMID: 37116494). The timescale used in the transcription assays shown in Fig. 1K is anomalously long (15-60 min), as the runoff reaction is completed within 2-3 min (see PMID: 30102406). Moreover, inactivation of Mtb RNAP after prolonged (>20 min) incubation at 37 °C and RNA degradation may bias the results.

We also carefully considered the suggested IC6 to IC7 extension experiment. However, primer-dependent reactions would generate RNA aligned in registers such

as -6 to +1, which neither reproduce the native +1 to +7 register formed during authentic de novo initiation nor correspond to the -7 to -1 register mentioned by the reviewer. Structural analysis further shows that the paused intermediate occurs during the 6 to 7 nt transition rather than at a single discrete RNA length. The 7 nt paused complex adopts a half-translocated architecture in which the RNA has reached 7 nt but the template DNA remains in a pre-translocated (6 nt equivalent) state, a well-established hallmark of paused RNAP intermediates (PNAS 2023, 120: e2218516120; Mol Cell 2023, 83:1474–1488). Thus, pausing intrinsically reflects the transition between IC6 and IC7, and cannot be fully captured by primer-initiated extension assays that bypass the scrunching-dependent formation of this intermediate. Moreover, using a fixed-length primer constrains RNAP to extend only from that preset register and prevents detection of shorter RNAs, including species that may populate earlier paused states. For this reason, we used a de novo initiation assay, which allows the enzyme to sample all early intermediates without imposing a predefined RNA length. Among the studies cited by the reviewer, rapid sub-minute kinetics were obtained either in *E. coli* de novo systems or in *Mtb* primer-initiated systems. Neither condition reflects de novo initiation by *Mtb* RNAP. In our hands, de novo transcription by *Mtb* RNAP on short promoter scaffolds produces very low levels of short RNAs within the sub-minute timescale, making a reliable kinetic comparison between IC6 and IC7 technically infeasible. In the revised manuscript, we also clarify in the Discussion (**Page 23, Lines 454-469**) the rationale for relying on de novo initiation assays, which allow the complex to populate the full spectrum of early intermediates without imposing a fixed RNA register.

We thank the reviewer for raising the possibility of RNAP inactivation and RNA degradation during >20 min incubations. We carefully examined this concern in light of published data. Notably, in Mol Cell 2023, 83:1474–1488, *Mtb* RNAP was incubated for 15 min to form a halted complex and then subjected to an additional 15 min transcription reaction, with no indication of loss of enzymatic activity or RNA degradation during this ≥30-min period at 37 °C. This indicates that *Mtb* RNAP remains catalytically competent well beyond 20 min under standard reaction conditions. In our assays, all buffers and consumables were DEPC-treated and RNase-free, making RNA degradation negligible. This is also consistent with the Mol Cell 2023, 83:1474–1488 study, in which no degradation of short RNAs was observed over comparable timeframes.

Finally, even if one of the reported structures corresponds to a paused state, its physiological relevance remains unclear, since no pause can be detected at the

promoter template under de novo initiation. This limitation should be acknowledged in the Discussion.

We thank the reviewer for noting that one of our structures corresponds to a paused intermediate. Initial-transcription pausing is widely recognized as a physiologically relevant checkpoint (Mol Cell 2016, 63:939–950; Nat Commun 2018, 9:1478), and the scrunched intermediate naturally leads to both extension and abortive release. Thus, de novo assays cannot cleanly separate these outcomes at the 6–7 nt stage. This does not affect the interpretation that the paused architecture we resolve represents a functional intermediate. In our previous revision, we already clarified that pausing is not assigned to a strict RNA length of 6 or 7 nt, but occurs during the 6 to 7 nt transition, consistent with the half-translocated architecture we observe, where the RNA is at 7 nt while the template DNA remains in a 6 nt equivalent position. In the present revision, we have now explicitly added to the Discussion (**Page 22, Lines 445-454**) a description of the limitations of the de novo assay, together with an explanation of how the matched +1 to +7 scaffold and the pre-melted scaffold relate to each other and why the paused intermediate captured by our structures remains physiologically meaningful.

Consequently, the manuscript's discussions regarding pausing “checkpoints” and the effect of CarD on pausing are not warranted. Thus, CarD can perfectly well reduce abortive initiation and that will explain diminishing of the +6/+7 band.

Initial-transcription pausing is widely recognized as an early checkpoint in transcription initiation, as the scrunched intermediate can naturally proceed either toward productive extension or toward abortive release (Nucleic Acids Res, 2025, 53: gkaf857; Mol Cell, 2020, 79:797-811; Mol Cell 2016, 63: 939 – 950; Nat Commun 2018, 9: 1478). Thus, the checkpoint concept is well supported by current mechanistic understanding. Because abortive RNAs arise directly from this paused intermediate, the 6–7 nt species observed in our assays—whether classified as paused or abortive—reflect the same underlying early-initiation checkpoint. In this context, the reduction of these RNAs upon addition of CarD is consistent with CarD stabilizing the open transcription bubble and favoring progression into productive extension rather than lingering in the paused/abortive branch. This interpretation is fully supported by both our structural (**Fig. 4A-D**) and CarD kinetic data (**Fig. 4G**), which consistently show that CarD stabilizes the transcription bubble and thereby shortens the lifetime of the paused intermediate. To ensure clarity and rigor, we have also refined the wording throughout the Introduction (**Page 6, Lines 104-108**) and Results (**Page 19, Lines 365-367**) to avoid overstating the role of CarD in pausing. In the revised manuscript, CarD is described only in terms of functions that are supported by our data, such as

stabilizing the transcription bubble and influencing the balance between abortive and productive outcomes. We have avoided language that would imply a specific or exclusive effect of CarD on the paused state.

2. Incorrect calculation of FRET efficiency.

In the first round of review, I recommended removing the FRET data because they did not appear reliable. Examination of the revised Methods section has reinforced these concerns.

We thank the reviewer for this helpful recommendation. In the current revision, we have removed the entire FRET dataset as suggested. As noted previously, the ensemble FRET signals were subject to technical limitations that made their interpretation uncertain. Because these data were not essential for our mechanistic conclusions, their removal results in a cleaner and more rigorous presentation.

Two major issues are outlined below:

a) The authors state that they calculated FRET efficiency (eFRET) “based on the donor (Cy3, D) fluorescence intensity.” In principle, one may calculate E from donor fluorescence intensity (ID) if a donor-only control is used (in this case it should be DNA labeled with Cy3 only). Under those conditions, E can indeed be calculated as $E = 1 - (ID/ID_{\text{only}})$. However, according to the Methods section, this was not done. Instead, the authors appear to calculate a ratio between Cy3 fluorescence in RNAP-DNA complexes and Cy3 fluorescence in DNA-only samples, which is not a valid method.

If the authors wish to avoid using a donor-only control, they could calculate E from the acceptor (Cy5) fluorescence intensity (IA) using $E = IA / (ID + IA)$ (e.g., see PMID: 29806016).

We sincerely thank the reviewer for the careful analysis of our FRET calculation. We fully acknowledge the concern that the donor-only normalization procedure, as well as the interpretation of Cy3 quenching, requires rigorous controls to ensure quantitative accuracy. Since the FRET experiment was intended only as an auxiliary qualitative readout and is not essential for any of the mechanistic conclusions of the manuscript, we have removed the entire FRET dataset, including the Cy3/Cy5 fluorescence measurements, from the revised version. This revision resolves the concerns regarding the calculation of FRET efficiency.

b) Reliance solely on Cy3 fluorescence is inappropriate because Cy3 is strongly influenced by its local environment, including proximity to protein and nucleic acids (see PMID: 31235181). Therefore, the observed changes in Cy3 fluorescence under different conditions are not necessarily due to changes in FRET; they may simply reflect environment-induced modulation of Cy3 fluorescence. This concern is particularly relevant in the authors' experimental setup, where Cy3 is positioned in the single-stranded DNA region at position -10 relative to the transcription start site. Under these conditions, the 5' end of 8-9 nt RNA would be close to the dye and could alter its fluorescence, whereas 6-7 nt RNAs would be more distant and would not affect it in the same way.

We also appreciate the reviewer's point regarding the strong environmental dependence of Cy3 fluorescence and the possibility that longer (8–9 nt) RNAs could influence Cy3 intensity independently of FRET. We agree that such effects could complicate interpretation. Because these issues cannot be fully excluded and the dataset is not required to support our main conclusions, we have removed the FRET experiment and its interpretation from the revised manuscript. The removal of this dataset avoids potential confounding effects while keeping the mechanistic narrative fully supported by the structural and biochemical data.

We sincerely thank the reviewers and the editor for their thoughtful and constructive comments, which have helped us further improve the clarity and rigor of the manuscript. In this revised version, we have carefully addressed all remaining points raised by the review through focused clarification and minor textual revisions.

In this revision, we have:

- Added a brief citation and discussion of very recent work on the IC6 to IC7 transition and its branching into abortive and productive pathways, further supporting our interpretation of the paused intermediate.
- Clarified the interpretation of the run-off to pause ratio In vitro transcription assay by noting the influence of extended incubation time, and briefly commented on the identity of the intermediate band between the 6–7 nt species and the run-off product.

Referee's remarks are shown in black and our responses are in blue. All modifications in the revised manuscript are highlighted in red.

Reviewer #2 (Remarks to the Author):

The revised manuscript is much improved, and several key conclusions are now more clearly supported.

We thank the reviewer for this encouraging comment and are pleased that the revised manuscript more clearly supports our key conclusions.

Regarding the transition from IC6 to IC7, the authors' discussion is accurate, and their reference to associated literature is relevant and appropriate; the paused state is an intermediate that can then give rise to different paths; this path distribution may be different in *Mtb* vs *E.coli*, but the gels qualitatively support the presence of states that give rise to 6-mer (and 7-mer) RNAs, likely due to abortive initiation. I also note that there is very recent work (doi.org/10.1101/2025.11.27.690701) on this transition and the paths emanating from it, as well as the sequence dependence of pausing, which the authors may want to cite to further support their argument.

We thank the reviewer for the positive assessment that our discussion of the IC6 to IC7 transition is accurate and that the cited literature is relevant and appropriate. We are pleased that the reviewer agrees with our interpretation that the paused state represents a transient intermediate that can give rise to multiple downstream pathways, and that our gels qualitatively support the presence of states leading to 6-nt and 7-nt RNA species, likely reflecting abortive initiation from this checkpoint.

We thank the reviewer for pointing out this very recent preprint ([doi: 10.1101/2025.11.27.690701](https://doi.org/10.1101/2025.11.27.690701)). We agree that this study provides additional support for the view that the 6–7 nt transition represents a branching intermediate during initiation, with outcomes influenced by sequence context and kinetic partitioning. We have now cited this work (**Reference 65**) in the Discussion (**Page 22, Lines 431-433**) and briefly noted its consistency with our interpretation.

The removal of the FRET data is a positive change, as the previous analysis was convoluted, had many caveats and distracted from the main conclusions.

We thank the reviewer for this positive assessment. We agree that removal of the ensemble FRET data improves the clarity and focus of the manuscript, and we are pleased that the reviewer feels this change helps to better highlight the main structural and mechanistic conclusions.

The interpretations related to the premelted and fully double-stranded DNA substrates is also robust and strengthens the overall model.

We thank the reviewer for this encouraging assessment. We are pleased that the interpretation of the premelted and fully double-stranded DNA substrates is considered robust and that it is viewed as strengthening the overall mechanistic model.

The CarD effect (EDFig. 11) itself is clear and well supported. One point that warrants brief clarification is the high run-off to pause ratio (EDFig. 11). The long reaction time (30 min) may contribute to this effect, and a short discussion of possible causes would be helpful. The authors may also wish to comment on the identity of the major band between the run-off product and the 6 – 7 nt species.

We thank the reviewer for recognizing that the CarD effect is clear and well supported. We agree that the relatively high run-off to pause ratio is likely influenced by the long reaction time (30 min), which was chosen to allow sufficient accumulation of the transient 6–7 nt paused species for reliable detection. We now briefly note in the Discussion (**Page 23, Lines 455-458**) that the relatively long incubation time allows continued transcript extension and accumulation of longer RNA products, which can lead to an increased proportion of run-off RNA relative to short paused species, without altering the identification or stability of the 6–7 nt initiation-paused intermediate.

Regarding the prominent band between the run-off product and the 6–7 nt species, we believe this most likely corresponds to a short, prematurely terminated or transiently stalled elongation product formed during the early stages of transcription initiation. Importantly, its presence does not affect the identification of the 6–7 nt paused species or the interpretation of the CarD-dependent stabilization of the initiation-paused intermediate. We have added a short comment in the Discussion (**Page 24, Lines 466-469**) to acknowledge this band and to clarify that it does not alter our mechanistic conclusions.